

# Biomass burning fuel consumption dynamics in the (sub)tropics assessed from satellite

N. Andela[1,2], G. R. van der Werf[1], J. W. Kaiser[3], T. T. van Leeuwen[4,5], M. J. Wooster[6,7], and C. E. R. Lehmann[8]

[1]Faculty of Earth and Life Sciences, VU University, Amsterdam, The Netherlands
[2]Biospheric Sciences Laboratory, NASA Goddard Space Flight Center, Greenbelt, MD 20771, USA
[3]Max-Planck-Institut für Chemie, Mainz, Germany
[4]SRON Netherlands Institute for Space Research, Utrecht, The Netherlands
[5]Institute for Marine and Atmospheric Research Utrecht, Utrecht, The Netherlands
[6]Kings College London, Environmental Monitoring and Modelling Research Group, Department of Geography, London, WC2R 2LS, UK
[7]NERC National Centre for Earth Observation (NCEO), UK
[8]School of GeoSciences, University of Edinburgh, Edinburgh, EH9 3JN, UK

*Correspondence to*: N. Andela (niels.andela@nasa.gov)

**Abstract.** Landscape fires occur on a large scale in (sub)tropical savannas and grasslands, affecting ecosystem dynamics, regional air quality and concentrations of atmospheric trace gasses. Fuel consumption per unit of area burned is an important but poorly constrained parameter in fire emission modelling. We combined satellite-derived burned area with fire radiative power (FRP) data to derive fuel consumption estimates for land cover types with low tree cover in South America, Sub-Saharan Africa, and Australia. We developed a new approach to estimate fuel consumption, based on FRP data from the polar orbiting MODerate-resolution Imaging Spectroradiometer (MODIS) and the geostationary Spinning Enhanced Visible and Infrared Imager (SEVIRI) in combination with MODIS burned area estimates. The fuel consumption estimates based on the geostationary and polar orbiting instruments showed good agreement in terms of spatial patterns, but absolute fuel consumption estimates remained more uncertain. Fuel consumption varies considerably in space and time, complicating the comparison of various approaches and using field measurements to constrain our results. Spatial patterns in fuel consumption could be partly explained by vegetation productivity and fire return periods. In South America, most fires occurred in savannas with relatively long fire return periods, resulting in comparatively high fuel consumption as opposed to the more frequently burning savannas in Sub-Saharan Africa. Strikingly, we found the infrequently burning interior of Australia having higher fuel consumption than the more productive but frequently burning savannas in northern Australia. Vegetation type also played an important role in explaining the distribution of fuel consumption, both by affecting fuel build up rates and fire return periods. Hummock grasslands, which were responsible for a large share of Australian biomass burning, showed larger fuel build up rates than equally productive grasslands in Africa, although this effect might have been partially driven by the presence of grazers in Africa. Finally, land management in the form of deforestation and agriculture also considerably affected fuel consumption regionally. We conclude that combining FRP and burned area estimates, calibrated against field measurements, is a promising approach in deriving quantitative estimates of fuel consumption.



Satellite derived fuel consumption estimates may both challenge our current understanding of spatiotemporal fuel consumption dynamics and serve as reference datasets to improve biogeochemical modelling approaches. Future field studies especially designed to validate satellite-based products, or airborne remote sensing, may further improve confidence in the absolute fuel consumption estimates which are quickly becoming the weakest link in fire emissions estimates.

## 5  1 Introduction

Landscape fires play an important role in many ecosystems across the globe, with (sub)tropical savannas of intermediate productivity being most frequently burned (Bowman et al., 2009). Within those (sub)tropical ecosystems, humans are responsible for most of the ignitions and fires have been actively managed for thousands of years (Stott, 2000; Archibald et al., 2010, 2011), partly aided by the vegetation traits of these regions which make them inherently flammable (Archibald et
al., 2009). Landscape fires promote open canopy grassy vegetation over closed canopy woody vegetation (Scholes and Archer, 1997; Bond et al., 2005; Lehmann et al., 2011), providing competitive advantages to grassy rather than woody species in frequently burning landscapes (Sankaran et al., 2008). This fire driven tree – grass competition is further affected by the occurrence of different vegetation traits on the continents (Staver et al., 2011; Lehmann et al., 2014; Moncrieff et al., 2014). Due to the large scale at which biomass burning occurs, inter-annual variability in landscape fires is directly related to
(greenhouse) gas concentrations in the atmosphere (Langenfelds et al., 2002) and affects regional air quality (Crutzen et al., 1979; Langmann et al., 2009; Turquety et al., 2009; Aouizerats et al., 2015). Fire regimes and fire management vary widely across (sub)tropical regions (Archibald et al., 2013), while ongoing socio-economic developments are expected to increasingly affect landscape fires and vegetation patterns during the coming century (Chen et al., 2013; Grégoire et al., 2013; Andela and van der Werf, 2014). Fuel consumption per unit area burned (kg m$^{-2}$), hereafter called fuel consumption
for brevity, is a key parameter to better understand the consequences of changing management practices, vegetation characteristics and climate on fire regimes, or to estimate fire emissions. Yet, spatiotemporal dynamics of fuel consumption on a continental scale remain poorly understood (van Leeuwen et al., 2014).

With global annual burned area exceeding the size of India (Giglio et al., 2013) or even the European Union (Randerson et
al., 2012), satellite remote sensing is an important source of data to understand the spatiotemporal dynamics of fire. Over the last decade, several new satellite observing systems have become operational, greatly improving our understanding of fire dynamics and fire emission estimates. For example, vegetation productivity (Running et al., 2004) and fire return periods (Archibald et al., 2013) can now be estimated using satellite imagery and data. Broadly speaking, two types of satellite datasets are available to study fire dynamics. These are satellite-derived estimates of burned area that are based on changes
in surface reflectance over time (Giglio et al., 2006b, 2013) and active fire observations often accompanied by information on fire radiative power (FRP; Giglio et al., 2006a; Roberts and Wooster, 2008).



Both data types have advantages and disadvantages for the purpose of estimating fire emissions. Burned area remains visible for several days to months after the fire occurred, allowing observations of fires that were obscured by clouds during the satellite overpass, as long as cloud cover is not too persistent (Roy et al., 2008). However, small fires are generally not detected by the burned area algorithm (Randerson et al., 2012) and fuel consumption has to be modelled in case burned area

is used to calculate emissions (van der Werf et al., 2010). Active fire observations on the other hand often include FRP associated with the detected fire, which can be used to estimate fire radiative energy (FRE) which is directly related to dry matter burned (Wooster et al., 2005). When FRP data of geostationary instruments are used, the full fire diurnal cycle is observed and FRE and dry matter burned can be estimated by integrating the FRP observations over time (Roberts et al., 2005). However, geostationary satellites are located relatively far from the Earth and therefore have a relatively coarse pixel

size. Consequently the smallest fires with low FRP often fall below their detection threshold (Freeborn et al., 2009). Polar orbiting satellites, like the MODerate-resolution Imaging Spectroradiometer (MODIS) are located closer to the Earth and therefore have a higher spatial resolution and sensitivity to small fires. However, with approximately four daily observations under ideal conditions the MODIS instruments provide relatively poor sampling of the fire diurnal cycle (Ellicott et al., 2009; Vermote et al., 2009; Freeborn et al., 2011). On top of the orbit-specific limitations, active fire observations from both

polar orbiting and geostationary instruments are sensitive to cloud cover, and radiation of surface fires may be partly obscured by tree cover (Freeborn et al., 2014a).

To date, most knowledge on fuel consumption dynamics stems from a limited number of field campaigns (summarized in van Leeuwen et al., 2014). These studies provide great detail and have considerably advanced our understanding of fuel

consumption dynamics, but upscaling is problematic because fuel consumption is highly variable in space and time (Hoffa et al., 1999; Hély, 2003; Boschetti and Roy, 2009). As an alternative approach, Roberts et al. (2011) combined burned area data from MODIS with FRE estimated from the geostationary Meteosat Spinning Enhanced Visible and Infrared Imager (SEVIRI) instrument, creating the first fully satellite-derived fuel consumption estimate for Africa. Although at that time only one year of SEVIRI data was available, Roberts et al. (2011) found some striking differences between their estimates of

fuel consumption and the ones resulting from the biogeochemical modelling framework used in the Global Fire Emission Database version 3 (GFED3; van der Werf et al., 2010). The satellite-derived fuel consumption estimates of Roberts et al. (2011) were considerably lower than the ones from GFED in savanna regions, which may partly be explained by the low sensitivity of the SEVIRI instrument to small fires and an overestimate by GFED3. However, spatial patterns were also different, indicating that such methods can provide new insights in the distribution of fuel consumption and of fuel build up

processes.

The objective of this study is to gain further insights into the spatial distribution and drivers of fuel consumption. Initially, fuel consumption is estimated across Sub-Saharan Africa, by building upon the previous work of Roberts et al. (2009, 2011) that combined active fire FRP data from the geostationary SEVIRI instrument with burned area data from MODIS. We used





a similar approach, but with longer time series (2010 – 2014) in an attempt to provide more statistically representative fuel consumption estimates, for example in less frequently burning grid cells. Then we used a similar method but now based on MODIS FRP data to expand our study region to include South America and Australia, in addition to Sub-Saharan Africa. In situ fuel consumption observations were used to calibrate the MODIS-derived fuel consumption estimates to match field-measured values. Because FRP observations may be partly obscured by tree cover (Freeborn et al., 2014a), we limited our study to low tree cover land cover classes. Results were compared to the SEVIRI-derived fuel consumption estimates, and that derived using the biochemical GFED modelling framework. Finally, we used the spatial distribution of our fuel consumption estimates to explore the drivers of fuel consumption in the study regions.

## 2 Data

In this study we combined burned area data (Sect. 2.1) with FRP data to derive fuel consumption estimates. We commenced by following the approach of Roberts et al. (2011), based on FRP data provided by the geostationary SEVIRI instrument (Sect. 2.2), and then developed a new method using the FRP data from the polar orbiting MODIS instruments (Sect. 2.3). Information on land cover type, fire return periods and net primary productivity (NPP) were used to better understand the spatial variation in fuel consumption while results were also compared to fuel consumption estimates extracted from the GFED4s dataset (Sect. 2.4). Both methods to derive fuel consumption were based on the native resolution of the FRP data, but end results were rescaled to 0.25° resolution, for comparison to drivers and in case of the MODIS FRP-detections to include a representative sample size.

### 2.1 MODIS burned area

The MCD64A1 burned area dataset, based on land surface spectral reflectance observations made by the MODIS instruments aboard the Terra and Aqua satellites, provides daily 500 m resolution global burned area estimates from August 2000 onwards (Giglio et al., 2009, 2013). Partly because of the relatively high spatial resolution of the MODIS instruments, MODIS based burned area data was found to perform best out of several burned area products (Roy and Boschetti, 2009; Padilla et al., 2014). Despite this relatively good performance, the burned area product still often misses the smallest fires, which according to Randerson et al. (2012) may comprise a large fraction (over one third) of the overall global burned area.

We estimated the mean fire return period based on the 14 years of MCD64A1 burned area data, by recording how many times each 500 m resolution MODIS grid cell had burned during the 2001 – 2014 period and then dividing this by the 14 years. This method yields best results for frequently burning grid cells where the accuracy is thought to be high. For grid cells burning only once during the 14 years it is likely that in many cases the actual fire return period may in fact be longer, leading to an overestimation of fire return periods. For grid cells without any burned area observations no fire return period





could be calculated. We then calculated the mean fire return period for each 0.25° grid cell as the mean return period of all 500 m grid cells within each 0.25° grid cell, weighted by burned area.

## 2.2 SEVIRI FRP data

The SEVIRI instrument, aboard the geostationary Meteosat Second Generation satellites is located at 0° longitude and
latitude and provides active fire observations at 3 km spatial resolution at nadir, degrading with increasing view angle (Roberts and Wooster, 2008; Freeborn et al., 2011). The sensitivity of the instrument to small fires is lower than the sensitivity of the MODIS instruments due to the coarser pixel size, but the instrument provides 15 min interval observations capturing almost the full fire diurnal cycle, cloud cover permitting. Here we used the Meteosat SEVIRI FRP-PIXEL product providing FRP data at 15 min interval on the original SEVIRI spatial resolution (Roberts and Wooster, 2008; Wooster et al.,
2015). The FRP-PIXEL product is freely available and can be downloaded from the Land Surface Analysis Satellite Applications Facility (http://landsaf.meteo.pt), from the EUMETSAT EO Portal (https://eoportal.eumetsat.int/) or via the EUMETCAST dissemination service  (http://www.eumetsat.int), both in real-time and archived form.

## 2.3 MODIS FRP data

The MODIS instruments aboard the polar orbiting Terra (MOD) and Aqua (MYD) satellites provide global FRP data at 1 km
resolution (Giglio et al., 2006a). To calculate fuel consumption we used the MOD14 and MYD14 version 005 active fire data for the full period that both satellites were in orbit (2003 – 2014). The 1 km resolution translates into a higher sensitivity to small fires (i.e., low FRP), although the FRP sensitivity of MODIS decreases towards the swath edges (Freeborn et al., 2011). Because of the importance of a large sample size for our analysis, we included the MODIS FRP data from all active fire detections (low to high confidence active fire pixels).

The number of daily overpasses of the MODIS instruments is lowest in the tropics and increases towards the poles due to orbital convergence. Cloud cover permitting, the two MODIS instruments provide around four daily observations in the (sub)tropics. We combined information from the MOD03 and MYD03 geolocation datasets associated with each MODIS overpass with the MOD14 and MYD14 cloud cover data, to derive the mean daily MODIS detection opportunity (i.e., cloud
free overpasses) during the burning season in a similar way to the processing used in the Global Fire Assimilation System (GFAS; Kaiser et al., 2012). Because of the large size of the MOD03 and MYD03 data we based this part of the analysis on 4 years of data (2009 – 2012), enough to calculate a representative mean value. MODIS data are freely available and can be downloaded from NASA at http://reverb.echo.nasa.gov.

## 2.4 Other datasets

We derived information on land cover type from the MODIS MCD12C1 version 051 product, using the University of Maryland land cover classification (Friedl et al., 2002). In this study we focussed on low tree cover vegetation types



including savannas, woody savannas, grasslands, shrublands and croplands. Forests and bare or sparsely vegetated areas were excluded. Because the land cover fraction having closed shrublands was small and contained very little of the overall fire activity, we merged open and closed shrublands into one 'shrubland' class. The dominant land cover type was based on 2003 – 2012 data, because post 2012 data was not available to us at the time of the study.

Net Primary Production (NPP) was derived from the Terra MODIS MOD17A3 version 055 1 km annual product (Running et al., 2004), and we used the mean NPP over 2003 – 2010 (post 2010 data was not available). Units of NPP were in g C m$^{-2}$ yr$^{-1}$, and for comparison to estimates of fuel consumption in units of dry matter (DM) burned per m$^2$ we assumed a vegetation (fuel) carbon content of 45% (Andreae and Merlet, 2001; Barbosa and Fearnside, 2005).

Fuel consumption estimates from field studies were used to calibrate and evaluate the fuel consumption estimates from satellite. Peer reviewed studies were compiled into a field observation database for several biomes by van Leeuwen et al. (2014), and here we used their values for the savanna biome, including grasslands and (woody) savannas.

Finally, we compared our results to modelled fuel consumption estimates over the same period (2003 – 2014) extracted from the GFED4s dataset (0.25° spatial resolution). Methods used in GFED4s are based on GFED3.1 (van der Werf et al., 2010) but with two main improvements. The first one is the inclusion of small fire burned area (Randerson et al., 2012), the second one is further tuning of the model to better match fuel consumption estimates from the database of van Leeuwen et al. (2014). This involved mostly faster turnover rates of leaf and litter in the model to lower fuel consumption rates in low
treecover regions. GFED data can be downloaded from http://globalfiredata.org/.

**3 Methods**

The primary objective of this study was to provide further insights into the spatial distribution of vegetation fire fuel consumption in key (sub)tropical biomass burning regions, and also to provide insights into its most important drivers. We first derived a fuel consumption estimate for Sub-Saharan Africa using SEVIRI FRP data and the MCD64A1 burned area product, using an approach similar to that of Roberts et al. (2011). We derived FRE from the SEVIRI FRP data, which was
subsequently converted to an estimate of DM-burned using the conversion factor of Wooster et al. (2005), see Sect. 3.1. To expand our understanding of fuel consumption beyond Africa, we explored if a similar approach could be applied to MODIS FRP data. This approach was similar to the methods of Kaiser et al. (2012) but with a few adjustments to calculate fuel consumption. Rather than using a conversion factor based on laboratory experiments as in Wooster et al. (2005), we related
the FRE to in situ field observations to estimate fuel consumption (Sect. 3.2). We present results of this processing for three (sub)tropical biomass burning regions: South America, Sub-Saharan Africa and Australia. In those regions we explored the



potential drivers of the spatial distribution of fuel consumption. Finally, the results were compared to model-derived fuel consumption estimates of GFED4s.

## 3.1 Converting SEVIRI FRP to fuel consumption using laboratory measurements

Roberts et al. (2011) combined estimates of dry matter burned (kg), based on one year of FRP data from the geostationary Meteosat SEVIRI instrument, with MODIS-derived burned area mapping (m$^2$) to derive fuel consumption estimates (kg m$^{-2}$) for Africa. Here we followed a similar approach, but now we included five years of SEVIRI data (2010 – 2014), to get a better understanding of fuel consumption in infrequently burning zones, and to derive more representative mean fuel consumption estimates in general. An overview of this method is given in the flow chart of Fig. 1a, and explained in more detail below.

First, the daily burned area data (500 m resolution) were reprojected to the native SEVIRI imaging grid (3 km resolution at nadir). Because of the uncertainty of the burn date in the burned area product (Boschetti et al., 2010; Giglio et al., 2013), and the fact that a fire can burn multiple days, we followed Roberts et al. (2011) and assumed that all FRP detections within one week of the burned area observations (before or after; i.e., in total 15 days) in a given grid cell belonged to the same fire. Grid cells having only burned area observations but no corresponding FRP detections are likely related to fires having relatively low FRP or those that were obscured by clouds (Roberts et al., 2011). These areas (3% of annual burned area) were excluded from our analysis. Moreover, about half (54%) of the burned area detections showed over 20% cloud cover and/or missing data during the 15 day accumulation period possibly reducing FRE estimates. We decided not to exclude these data to maintain as large a sample as possible but we investigated the impact of this effect via a comparison of results including and excluding partial cloud cover and missing data.

As a second step, the 15 minute interval SEVIRI FRP detections were integrated over time to calculate FRE. This FRE was then converted into dry matter burned using the conversion factor (0.356 kg MJ$^{-1}$) based on lab experiments of various fuel types by Wooster et al. (2005). We limited the study to the spatial distribution of mean fuel consumption and calculated fuel consumption (*FC*) for each 0.25° grid cell (*x,y*) based on:

$$FC(SEVIRI)_{x,y} = \frac{\sum_{2010}^{2014} DM\_burned_{x,y}}{\sum_{2010}^{2014} BA_{x,y}} \tag{1}$$

Where $\sum_{2010}^{2014} DM\_burned$ corresponds to the sum of dry matter burned of each SEVIRI grid cell within the coarser 0.25° grid over the study period with a corresponding burned area observation, and $\sum_{2010}^{2014} BA$ is the sum of burned area (BA) for each 0.25° grid cell with corresponding FRE over the study period.



## 3.2 Converting MODIS FRP to fuel consumption using in situ measurements

With approximately four daily overpasses, MODIS provides only a sample of daily fire activity and FRP. Various approaches have been developed to derive FRE (Joules) and dry matter burned (kg) estimates from the MODIS FRP data (e.g., Ellicott et al., 2009; Freeborn et al., 2009, 2011; Vermote et al., 2009; Kaiser et al., 2012). However, methods to

convert MODIS FRP to FRE usually work at the relatively coarse spatial and/or temporal scale (e.g., 0.5° monthly) required to accumulate a statistically valid number of FRP observations. The sensitivity of the MODIS burned area product to 'small fires' is considerably worse than that of the MODIS active fire product (Randerson et al., 2012), and within each relatively large grid cell the proportion of FRP observations that originate from these small (unmapped) burned areas remains unknown. Therefore, these methods cannot directly be used to estimate fuel consumption. The method developed here to

derive FRE is similar to the one used within the GFAS version 1 (Heil et al., 2010; Kaiser et al., 2012), but observations of 'small fires' (having FRP detections but no corresponding burned area) were discarded (by working at the native MODIS 1 km resolution). Because the objective here was to estimate fuel consumption per unit area burned instead of total dry matter burned, the impact of ignoring the smallest fires is small, as long as fuel consumption in such fires is of a similar magnitude or their relative fraction is low. An overview of the method is shown in the flow chart of Fig. 1b, and explained in more

detail below.

As with the approach detailed in Sect. 3.1, the daily MODIS burned area data (500 m resolution) was rescaled to the resolution of the active fire product (for MODIS a 1 km resolution). Also, all MODIS FRP detections within a week before or a week after a 1 km grid cell was flagged as 'burned' were assumed to be part of the same fire. FRP detections without

corresponding burned area within this period were assumed to correspond to small fires and were excluded from the analysis. In contrast to the approach based on SEVIRI data, here all burned area observations were included. The FRP detections made by the MODIS instruments are more sensitive to small fires than the burned area product (Randerson et al., 2012), and it can therefore be reasonably assumed that the vast majority of fires that leave a detectable burned area signal will be observed by the MODIS instruments if there is a MODIS detection opportunity (i.e., a non-cloud obstructed overpass

of one of the MODIS instruments) during the fire.

The FRP recorded by the polar orbiting MODIS instruments are affected by the MODIS scan geometry (Freeborn et al., 2011), cloud cover, tree cover (Freeborn et al., 2014b), and the fire diurnal cycle and daily number and timing of overpasses (Andela et al., 2015). Hence, whilst a single MODIS FRP detection is somewhat representative of the overall fire activity in

a certain grid cell, its value is also influenced by these other factors (e.g., Boschetti and Roy, 2009; Freeborn et al., 2009; Andela et al., 2015). Moreover, temporal variations in fuel consumption may be considerable, driven by climate, vegetation type, management, and fire return periods (Shea et al., 1996; Hély, 2003; Savadogo et al., 2007). Minimising the impact of these type of perturbations is in part why methods developed to estimate FRE from MODIS FRP generally require the





accumulation of MODIS FRP observations over relatively coarse spatiotemporal scales (e.g., Freeborn et al., 2009; Vermote et al., 2009). We further investigated the combined effect of all these factors on the FRP data by studying the distribution of FRP-observations for a frequently burning grid cell in Africa.

Following the methods applied within GFAS (Heil et al., 2010; Kaiser et al., 2012), FRE was estimated by assuming that the observed daily fire activity (i.e., FRP) at cloud free MODIS overpasses is representative for daily fire activity. To create a sufficiently large and 'representative' sample size, burned area detections and FRP detections with corresponding burned area were aggregated to a 0.25° spatial resolution for the full period that both Aqua and Terra were in orbit (2003 – 2014). Subsequently the total emitted FRE (J) over the study period was calculated per grid cell as the sum of FRP (Watt or J s$^{-1}$)

multiplied by the mean duration between two MODIS detection opportunities (s) during the burning season (calculated using the mean number of cloud free overpasses per day weighted by monthly burned area). This way we implicitly correct for variation in the daily detection opportunity caused by cloud cover and/or the MODIS orbits (e.g., Kaiser et al., 2012; Andela et al., 2015). For further analysis we only include those 0.25° grid cells containing at least 50 MODIS FRP detections (together responsible for 96% of annual burned area).

Because of the uncertainties in the FRE estimates, we calibrated our results directly against field measurements. We used simple linear regression forced through the origin, to derive a conversion factor (kg MJ$^{-1}$) between the MODIS-derived FRE per unit area burned (MJ m$^{-2}$) and the corresponding field measurements of fuel consumption (kg m$^{-2}$) compiled by van Leeuwen et al. (2014). From this field measurement database we included all measurements conducted in grasslands,

savannas and woody savannas (Table 1). The results were also compared to the results based on the approach using the SEVIRI instrument outlined in Sect. 3.1, however in that case we did apply the conversion factor as suggested by Wooster et al. (2005) to enable direct comparison.





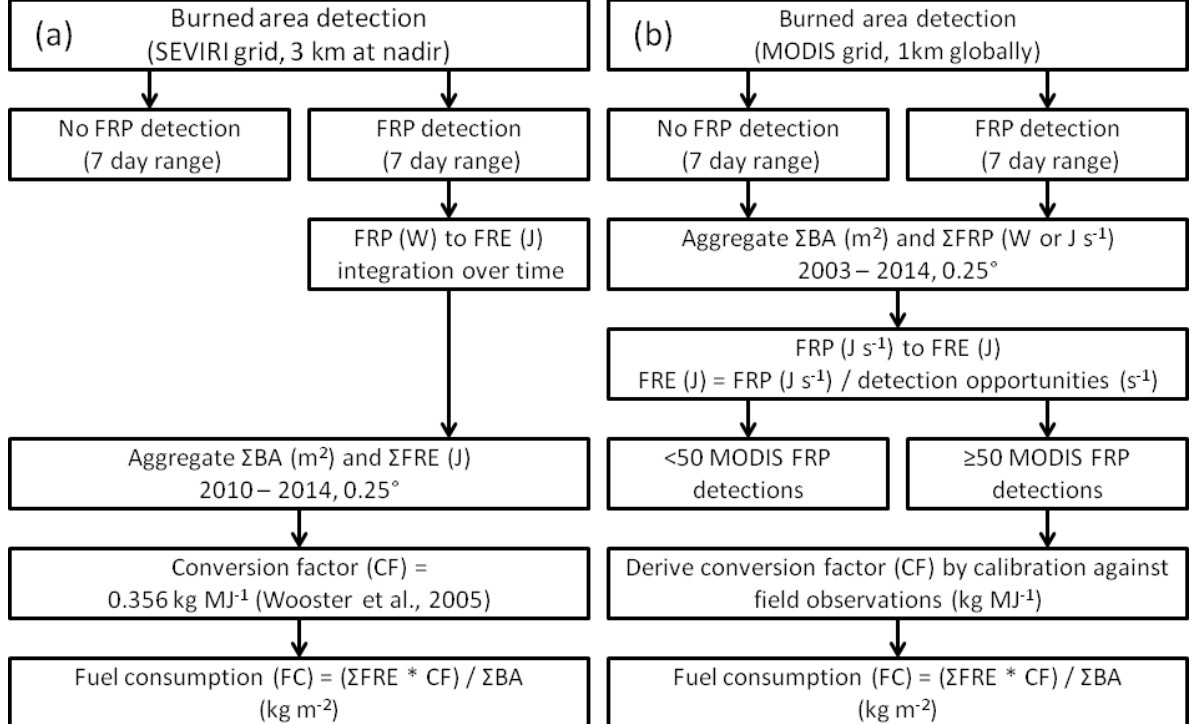

**Figure 1.** Methods to derive 0.25° fuel consumption estimates based on two different approaches. (**a**) The pathway used to combine FRP data of the geostationary SEVIRI instrument with burned area data to derive fuel consumption (Roberts et al., 2011; Sect. 3.1). (**b**) The pathway used to derive fuel consumption by combining FRP data of the polar orbiting MODIS instruments with burned area data (Sect. 3.2). Note that FRP detections without corresponding burned area, associated with small fires, are excluded in both processing chains.

## 4 Results

### 4.1 Comparing SEVIRI and MODIS-derived fuel consumption

To provide new insights in the specific qualities and limitations of polar orbiting and geostationary based FRP data, we compared the mean fuel consumption (kg m$^{-2}$) estimates based on our approach using SEVIRI FRP data (Fig. 2a) with our approach using MODIS FRP data (Fig. 2c). Although later on a new conversion factor is derived to convert the MODIS based FRE into DM-burned, here both FRE estimates were converted using the same conversion factor (0.356 kg MJ$^{-1}$; Wooster et al., 2005) for comparison. We used linear regression fitted through the origin (Fig. 2b), to compare the results. Total estimated FRE, and thus fuel consumption, based on the MODIS instruments was roughly two times larger than SEVIRI-derived fuel consumption. On top of these absolute differences, the spatial patterns were not uniform (Fig. 2b and d), for which we identified two main causes: first the MODIS based fuel consumption was consistently higher in south



eastern Africa (e.g., Mozambique and Madagascar), likely because of the decreasing sensitivity of the SEVIRI instrument at the greater off-nadir angle over this region (e.g., Freeborn et al., 2014b); and second the difference between both approaches was larger in areas of infrequent fires (compare Fig. 2d and Fig. A1a), possibly explained by the relatively short and slightly different periods of data availability (SEVIRI 2010 – 2014; MODIS 2003 – 2014) in combination with large inter-annual

5  variations in fuel consumption (e.g., Hély, 2003). To prevent these differences from affecting our estimated correlation too much, we only included frequently burning grid cells (burned area ≥ 15 % yr$^{-1}$) and those having a surface area of the SEVIRI FRP-PIXEL product grid cells below 12 km$^2$ (minimum value is 9 km$^2$ at nadir) during the linear regression shown in Fig. 2b. This resulted in reasonable correlation ($r^2 = 0.42$; n = 6621). Partial cloud cover and missing data were also affecting the analysis, and we found that 54% of the annual burned area occurred during periods of reduced data availability

10  (below 80%). When excluding these events, the absolute difference between MODIS and SEVIRI based fuel consumption became somewhat smaller (i.e., the slope in Fig. 2b became 0.61), demonstrating that periods of reduced observations were partly responsible for the underestimation in SEVIRI-derived fuel consumption. However, by excluding this 54% of the data, the correlation between MODIS and SEVIRI based fuel consumption was reduced ($r^2$=0.31), due to the heterogeneous nature of fuel consumption.

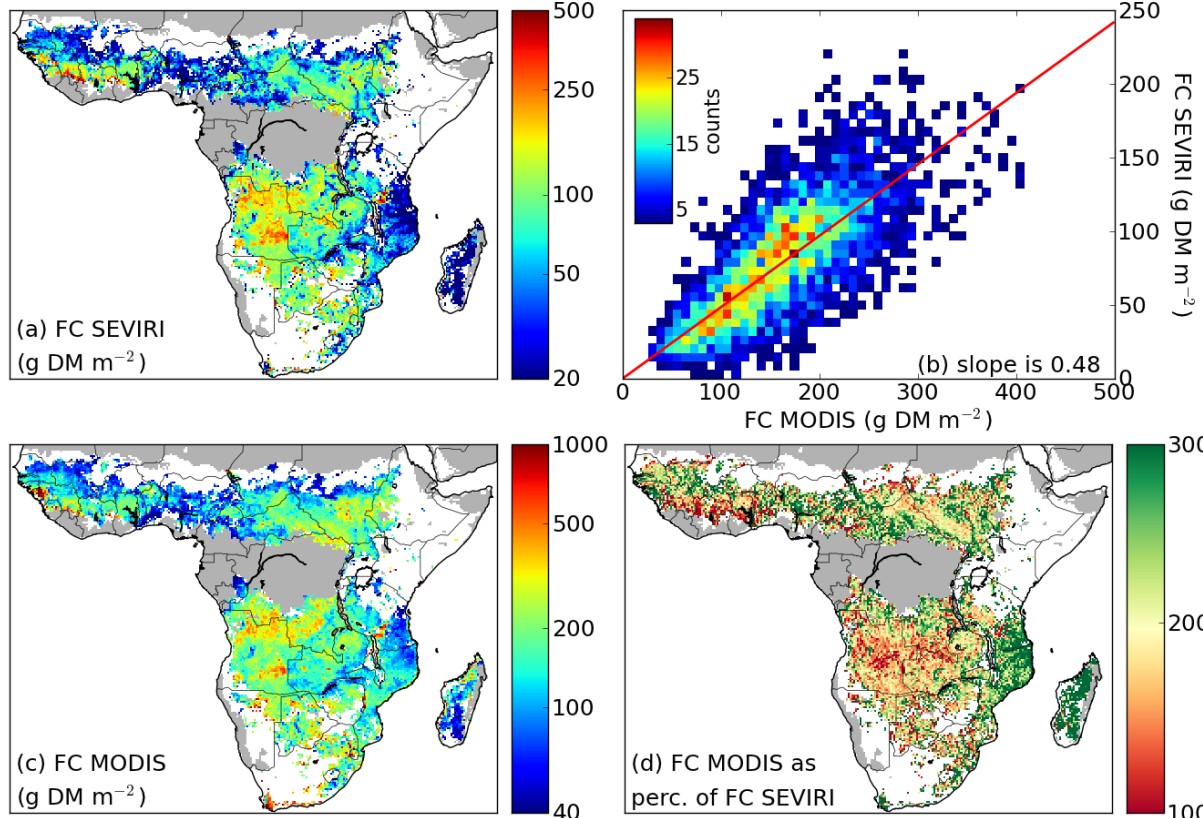

**Figure 2.** Comparison of fuel consumption (FC) estimates derived by combining FRP and burned area data. (**a**) Fuel consumption derived using SEVIRI FRP data. (**b**) Correlation between fuel consumption estimates based on the SEVIRI and



MODIS FRP data. (**c**) Fuel consumption derived using MODIS FRP data. (**d**) MODIS based fuel consumption estimates as percentage of the SEVIRI based estimates. In both cases FC was derived using the same 0.368 kg MJ$^{-1}$ conversion factor (Wooster et al., 2005). Grid cells with dominant land cover 'forest' or 'bare or sparsely vegetated' were excluded from our analysis and are masked grey, while water and grid cells with less than 50 MODIS FRP detections during our study period
(2003 – 2014) are shown in white in all figures.

## 4.2 Converting MODIS FRP to fuel consumption using in situ measurements

Although the FRE per unit area burned can be converted to fuel consumption using the conversion factor found by Wooster et al. (2005) during laboratory experiments which we have used so far, some instrument specific issues may further affect the FRE estimates from space (see methods). In order to correct for uncertainties in the MODIS derived FRE estimates, we
derived an alternative conversion factor by comparing the MODIS FRE per unit area burned directly to field measurements instead (Fig. 3a). This way, we found a conversion factor of 0.572 kg MJ$^{-1}$, about 1.5 times higher than the conversion factors of 0.368 kg MJ$^{-1}$ based on the laboratory experiments (Wooster et al., 2005). Due to the limited number of field observations and a number of outliers the coefficient of determination ($r^2$) is considered reasonable (0.41), something we return to in the discussion.

Figure 3b shows the distribution of MODIS FRP detections for a frequently burning 0.25° grid cell in northern Africa for the 2003 – 2014 period. As discussed in the methods, a single MODIS FRP detection is often not representative for the actual fuel consumption rate or fire activity, and it is more reasonably to take a representative sample (we used a minimum of 50 active fire pixel detections). For this particular 0.25° grid cell (Fig. 3b), over the full period there were 967 MODIS FRP
detections, having a sum of 39.7 GW, while total burned area was 5.7x10$^9$ m$^2$. During the burning season, the two MODIS instruments together observed the grid cell 2.8 times a day on average. The estimated FRE per unit area burned was therefore 0.22 MJ m$^{-2}$. Using the conversion factor derived below (Fig. 3a), the estimated fuel consumption for the grid cell shown in Fig. 3b is 124 g DM m$^{-2}$. To put this value in context, for this grid cell the mean NPP was 732 g DM m$^{-2}$ yr$^{-1}$ and the mean fire return period 1.75 years over the study period.



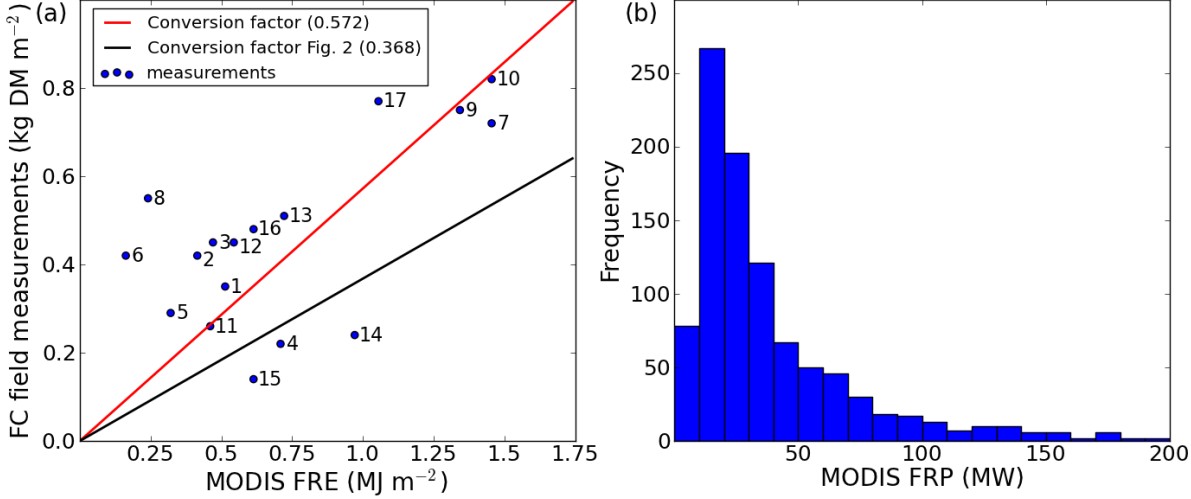

**Figure 3.** Relationship between MODIS FRP detections and fuel consumption (FC). (**a**) Best estimated conversion factor between MODIS derived FRE per unit area burned and field measurements of fuel consumption. The red line indicates the conversion factor used here and the black line the conversion factor found by Wooster et al. (2005) during laboratory experiments. The blue dots and numbers refer to the individual field studies (van Leeuwen et al., 2014; Table 1). (**b**) Distribution of MODIS FRP detections (binned in classes of 10 MW wide) for a frequently burning 0.25° grid cell in northern Africa (10.00 – 10.25° N, 24.00 – 24.25° E).

Table 1 provides an overview of the field studies used to calibrate the MODIS based fuel consumption estimate (Fig. 3a), and corresponding 0.25° fuel consumption estimates based on MODIS FRP detections. Most fuel consumption estimates based on field measurements are similar in magnitude to the ones derived here, although there are a few prominent outliers (Ref. 6, 8, 14 and 15). The field studies corresponding to Ref. 15 and 16 were carried out within the same 0.25° grid cell and illustrate that individual case studies are not always directly comparable with our 0.25° fuel consumption estimates due to the large spatial heterogeneity of fuel consumption.





**Table 1.** Fuel consumption estimates for grasslands, savannas and woody savannas, based on field studies compiled by van Leeuwen et al. (2014) and the corresponding 0.25° fuel consumption estimates derived here. For the field studies, numbers in parentheses show the standard deviation. N is the number of active fire detections by MODIS (2003 – 2014) for each 0.25° grid cell. References: (1,2,3) Shea et al. (1996)/Ward et al. (1996); (4) Hoffa et al. (1999); (5) Hély et al. (2003); (6) Savadogo et al. (2007); (7) Ward et al. (1992); (8) Bilbao and Medina (1996); (9) Miranda et al. (1996); (10) De Castro and Kauffman (1998); (11) Barbosa and Fearnside (2005); (12) Cook et al. (1994); (13) Hurst et al. (1994); (14) Rossiter-Rachor et al. (2008); (15) Russell-Smith et al. (2009); (16) Meyer et al. (2012); (17) Prasad et al. (2001).

| Ref. | Lat. | Lon. | FC (g DM m$^{-2}$) Field study | FC (g DM m$^{-2}$) MODIS | N | Description |
|------|------|------|------|------|------|-------------|
| 1 | 25.15 S | 31.14 E | 350 (140) | 294 | 226 | Lowveld sour bushveld savanna |
| 2 | 12.35 S | 30.21 E | 420 (100) | 237 | 1487 | Dambo, miombo, chitemene |
| 3 | 16.60 S | 27.15 E | 450 (–) | 269 | 216 | Semi-arid miombo |
| 4 | 14.52 S | 24.49 E | 220 (120) | 405 | 880 | Dambo and miombo |
| 5 | 15.00 S | 23.00 E | 290 (90) | 183 | 407 | Dambo and floodplain |
| 6 | 12.22N | 2.70 W | 420 (70) | 92 | 177 | Grazing and no grazing |
| 7 | 15.84 S | 47.95 W | 720 (90) | 832 | 126 | Different types of cerrado |
| 8 | 8.56 N | 67.25 W | 550 (190) | 138 | 232 | Protected savanna for 27 years |
| 9 | 15.51 S | 47.53 W | 750 (–) | 768 | 69 | Campo limpo and campo sujo |
| 10 | 15.84 S | 47.95 W | 820 (280) | 832 | 126 | Different types of cerrado |
| 11 | 3.75 N | 60.50 W | 260 (90) | 263 | 35 | Different types of cerrado |
| 12 | 12.40 S | 132.50 E | 450 (130) | 311 | 1885 | Woodland |
| 13 | 12.30 S | 133.00 E | 510 (–) | 413 | 1277 | Tropical savanna |
| 14 | 12.43 S | 131.49 E | 240 (110) | 555 | 433 | Grass and woody litter |
| 15 | 12.38 S | 133.55 E | 140 (160) | 351 | 1357 | Early and late season fires |
| 16 | 12.38 S | 133.55 E | 480 (–) | 351 | 1357 | Grass and open woodland |
| 17 | 17.65N | 81.75 E | 770 (260) | 603 | 20 | Woodland |

Fuel consumption for the three study regions was derived by applying the conversion factor (Fig. 3a; 0.572 kg MJ$^{-1}$) to the FRE per unit area burned (MJ m$^{-2}$) as estimated using the MODIS FRP-detections (Fig. 4). South America generally showed relatively high fuel consumption, with the fringes of the deforestation areas having by far the highest values (Fig. 4a). Sub-Saharan Africa has relatively low fuel consumption compared to Australia and southern America, with lowest fuel consumption found in East Africa and agricultural regions in western Africa (e.g., Nigeria; cf. Fig. 4b and Fig. A1h). Australia shows a surprising pattern where fuel consumption according to our approach in frequently burning savannas in



northern Australia appears to be lower than fuel consumption in the drier interior (Fig. 4c and Fig. A1c). The same pattern is observed in some arid regions of southern Africa where fires have long return periods (e.g., Namibia; Fig. 4b and Fig. A1b).

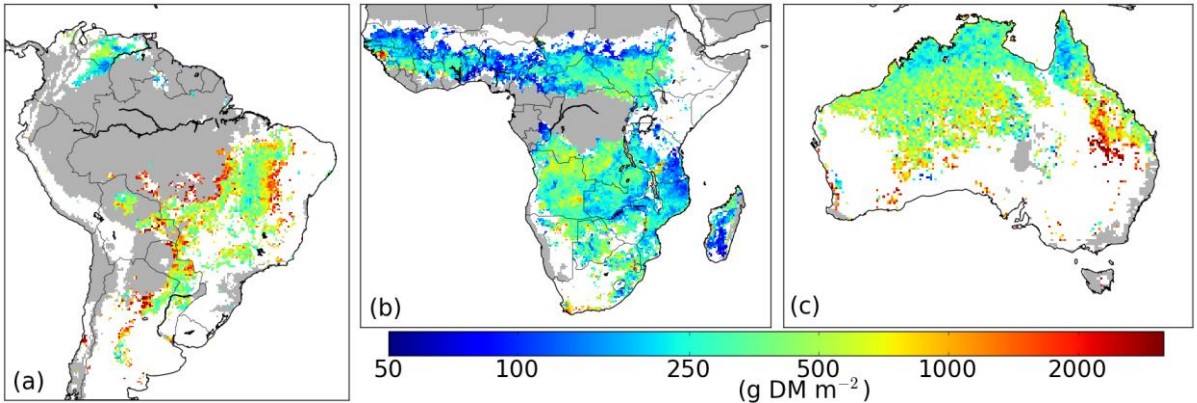

5 **Figure 4.** Distribution of fuel consumption based on MODIS FRP detections and the conversion factor derived from field studies. (**a**) South America, (**b**) Sub-Saharan Africa and (**c**) Australia.

### 4.3 Drivers and dynamics of fuel consumption

For each continent we assessed whether most fires occurred in productive or low productivity systems, and whether short or long fire return periods were most common (Fig. 5a – c). Then we explored the distribution of fuel consumption as a 10 function of productivity and fire return periods (Fig. 5d – f), followed by the possible role of land cover type in explaining these patterns (Fig. 5g – i). We found that biomass burning on the three continents occurred under very different conditions in terms of productivity and fire return periods. Within the South American study region most fires occurred in relatively productive savannas (NPP of 800 – 1600 g DM m$^{-2}$ yr$^{-1}$) and were characterized by relatively long fire return periods (3 – 8 yrs). Fuel consumption in this region was higher than under similar conditions (in terms of NPP and fire return period) in 15 Sub-Saharan Africa and Australia. African biomass burning was dominated by (woody) savanna fires of annual and biennial return periods which were observed over a wide range of NPP (500 – 2000 g DM m$^{-2}$ yr$^{-1}$). For the lower productivity African savannas and grasslands, we did not find large differences in fuel consumption between savannas that burn annually or biennially, and savannas with somewhat longer return periods (3 – 8 yrs). Only African savannas with fire return periods above 8 years showed again a somewhat higher fuel consumption. Strikingly, in the more productive African savannas fuel 20 consumption declined with longer fire return periods.

In Australia most burned area occurred in the savannas of intermediate productivity (500 – 1200 g DM m$^{-2}$ yr$^{-1}$) and low productivity Hummock grasslands (<500 g DM m$^{-2}$ yr$^{-1}$; Australian Native Vegetation Assessment, 2001), that were classified as shrublands by the MODIS land cover dataset. While in Sub-Saharan Africa most fires in the lower productivity 25 regions were fuelled by grasses that form well connected fuel beds, in Australia most fires occurred in poorly connected




Hummock grasslands that functionally act like shrublands. Both in Sub-Saharan Africa and in Australia regions classified as shrubs faced longer fire return periods than grasslands and savannas, but eventually burned with higher fuel consumption. But even when productivity and fire return periods were similar the fuel consumption in the low productivity (<500 g DM m$^{-2}$ yr$^{-1}$) Hummock grasslands of Australia was consistently higher than fuel consumption of the low productivity grasslands

5    in Sub-Saharan Africa.

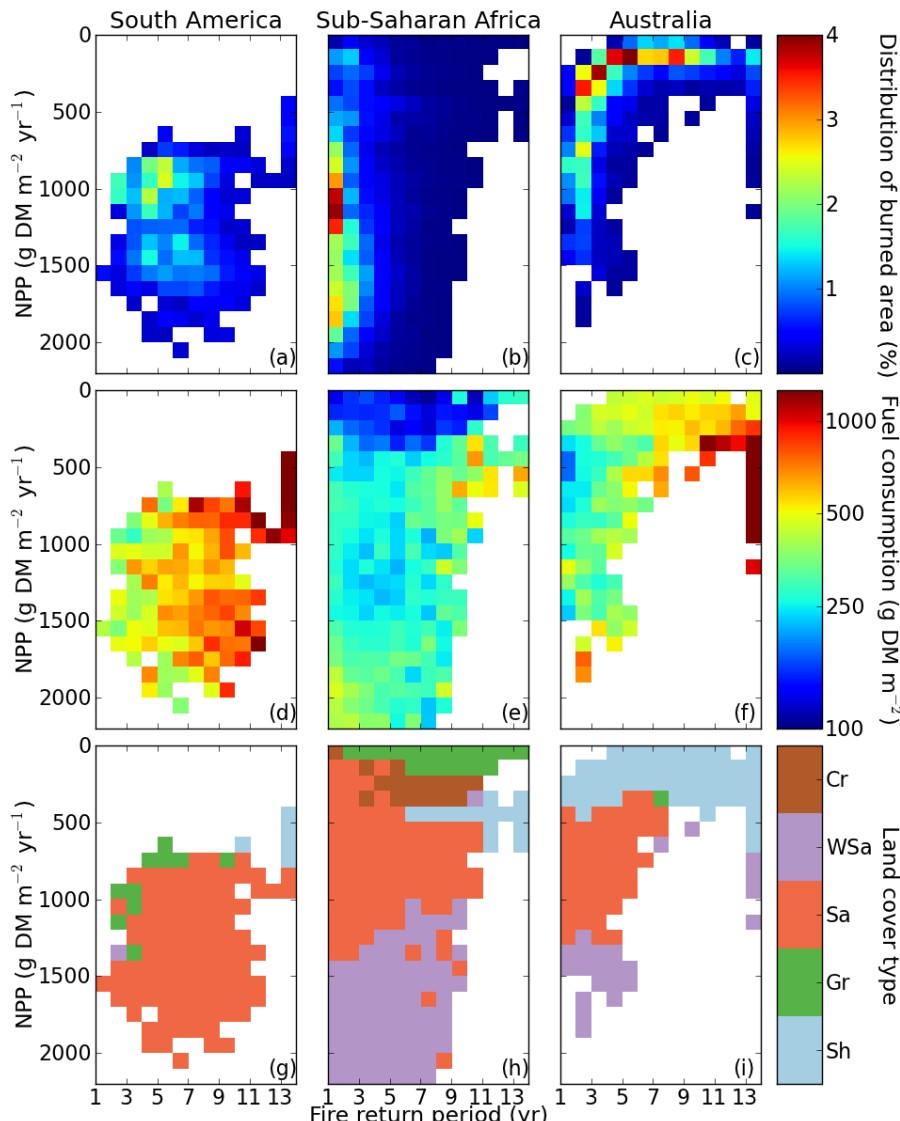

**Figure 5.** Distribution of burned area (**a-c**), fuel consumption (**d-f**) and dominant land cover type (**g-i**), all binned as a function of fire return periods and net primary productivity (NPP) for the three study regions (South America, Sub-Saharan Africa and Australia). Bins with burned area below 500 km$^2$ yr$^{-1}$ are masked in white. Abbreviations of land cover type stand

10   for: cropland (Cr), woody savanna (WSa), savanna (Sa), grassland (Gr) and shrubland (Sh).




Finally, we compared the fuel consumption estimates derived from the MODIS FRP-detections (Fig. 4) with fuel consumption estimates of GFED4s. Considerable differences were found between the two approaches (Fig. 6). The fuel consumption estimates derived here resulted in higher fuel consumption estimates for areas of lower productivity, especially

those areas dominated by shrublands; while fuel consumption estimates of GFED4s were generally higher in woody savannas, with higher productivity. Interestingly, the best comparison was found in zones of most frequent fire and short fire return periods (compare Fig. 6 with Fig. A1a – c).

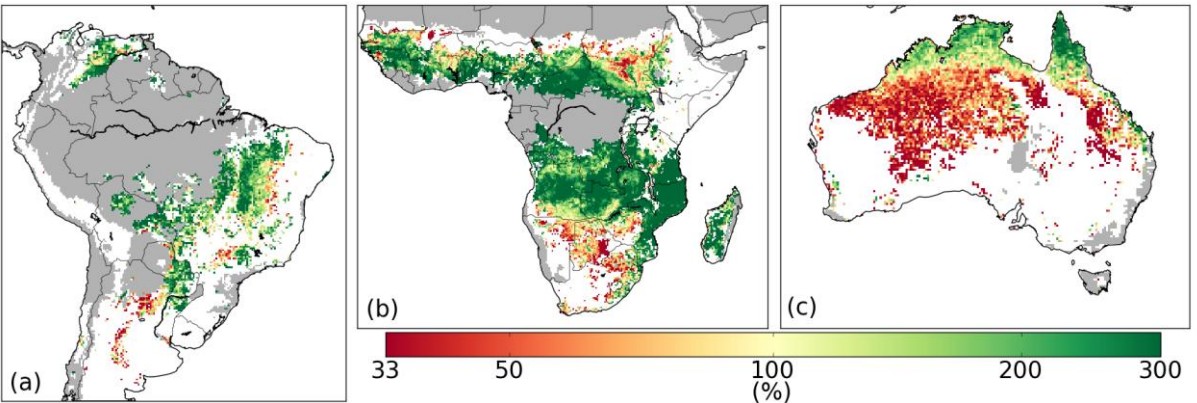

**Figure 6.** Fuel consumption estimated by GFED4s as a percentage of fuel consumption derived here (based on MODIS derived FRE, calibrated against in situ measurements). (**a**) South America, (**b**) Sub-Saharan Africa and (**c**) Australia.

## 5 Discussion

Understanding the global distribution of fuel consumption and fuel build up mechanisms is important to make landscape management decisions, understand the implications of changes in climate or vegetation patterns on fire dynamics, and to

derive accurate fire emission estimates. Boschetti et al. (2009) and Roberts et al. (2009, 2011) showed that fuel consumption per unit burned area estimates can be derived from combining burned area and active fire satellite products. Here we build upon their approaches and derived fuel consumption estimates for regions of low tree cover in South America, Sub-Saharan Africa and Australia, and explored the drivers of the spatial distribution. Following previous studies, we found that fuel consumption is highly heterogeneous (e.g., Hély, 2003; Savadogo et al., 2007; Boschetti and Roy, 2009; Roberts et al.,

2011). Consequently, obtaining representative field measurements is labour intensive, and only a limited number of studies have been carried out (van Leeuwen et al., 2014). Satellite-derived estimates of the spatial distribution of fuel consumption can therefore form an important addition to the scarce field measurements and may guide future field campaigns.





Here we discuss the pros and cons of the fuel consumption estimates presented in this paper and the current challenges for such an exercise (Sect. 5.1). We then discuss the drivers of fuel consumption in the three study regions, and compare the results found here to fuel consumption estimates of the GFED4s data (Sect. 5.2).

## 5.1 Satellite-derived fuel consumption estimates

In this study we explored the distribution of fuel consumption beyond the geostationary position of the SEVIRI instrument, and developed a method based on FRP detections of the polar orbiting MODIS instruments. Both geostationary and polar orbiting instruments have advantages, the geostationary SEVIRI instrument observes the full fire diurnal cycle, while the polar orbiting MODIS instruments only provides observations at certain fixed hours of the day, potentially leading to structural errors in the FRE estimates (Ellicott et al., 2009; Vermote et al., 2009; Freeborn et al., 2011; Andela et al., 2015).

However, the sensitivity of the MODIS instruments to small fire is much larger than that of the SEVIRI instrument (Freeborn et al., 2014b). In order to get a better understanding of the implications of these differences we compared the fuel consumption estimates based on both platforms using the FRE-to-DM-burned conversion factor found during laboratory experiments (Wooster et al., 2005). At first sight, very similar spatial patterns were found using polar orbiting or geostationary data, (compare Fig. 2a and c), providing confidence in the spatial distribution of the fuel consumption

estimates. However, many differences were also present (Fig. 2d). We found that a large part of the differences could be attributed to: (i) the large natural temporal variation in fuel consumption combined with the different periods of data availability and (ii) the different sensors characteristics. After excluding grid cells at higher off-nadir angles of the SEVIRI instrument and of infrequent fire occurrence, we found an $r^2$ of 0.42 between both approaches. Other potential sources of spatial discrepancy, like the fire diurnal cycle, were smaller. Finally, a large structural difference was observed, and SEVIRI-

derived fuel consumption was about half of the MODIS-derived fuel consumption. Such structural differences likely occur due to the different sensitivities of the instruments (Freeborn et al., 2009, 2014b). As compared to the MODIS instruments, the SEVIRI instrument likely underestimates fire activity in areas where a relatively large fraction of fire activity falls below the detection threshold (e.g., small fires, or fires partly obscured by trees; as discussed by e.g., Roberts and Wooster, 2008; Freeborn et al., 2009, 2014b; Roberts et al., 2011). In our analysis a small part of the structural difference could also be

explained by the fact that we did not correct for cloud cover and/or missing data in the SEVIRI based FC estimates. Not surprisingly, the best comparison between both methods was found in areas of frequent fires (Fig. 2d and A1) where fires often spread over large areas (Archibald et al., 2013), and are thus likely to be well observed by both instruments.

When deriving fuel consumption estimates based on the SEVIRI instrument, Roberts et al. (2011) found fuel consumption

estimates around 3.5 times lower than modelled values of GFED3.1 (van der Werf et al., 2010). Other studies that calculated fire emissions using the SEVIRI instrument found similar low estimates compared to GFED (e.g., Roberts and Wooster, 2008). Following previous studies, we find that about half of this discrepancy can be attributed to SEVIRI failing to detect the more weakly burning, but highly numerous, smaller fires that ultimately are responsible for around half of the emitted





FRE (Freeborn et al., 2009, 2014b). However, other sensor specific aspects also affect the FRE estimates (see Methods). We therefore decided to relate the FRE per unit area burned based on the MODIS instruments directly to field observations to convert them to measures of fuel consumption. We used the fuel consumption database of van Leeuwen et al. (2014) and the recommendation therein to use linear regression rather than biome mean values to derive the conversion factor. The conversion factor found using the field observations (Fig. 3; CF = 0.572 kg MJ$^{-1}$) was still around 1.5 times larger than the one found by Wooster et al. (2005; CF = 0.368 kg MJ$^{-1}$). This may for a large part be explained by sensor specific limitations, for example the lower sensitivity of the MODIS instruments towards the swath edges (Freeborn et al., 2011). The fire diurnal cycle in combination with the timing of the MODIS overpasses may also have affected absolute FRE estimates and thus the conversion factor derived here (Andela et al., 2015). Directly applying the conversion factor found by Wooster et al. (2005) on the MODIS derived FRE per burned area as derived here would therefore lead to an underestimation of fuel consumption (Fig. 3a). However, the number of field measurements was limited and our calibration was strongly influenced by a few field studies in more productive savannas. The correlation of the field observations and the 0.25° long term average fuel consumption estimates derived here (Fig. 3a; r$^2$=0.41) was affected by various factors. Most importantly, fuel consumption is both spatially and temporally highly heterogeneous (e.g., Hoffa et al., 1999; Hély, 2003; Govender et al., 2006), so even in the case of accurate fuel consumption estimates from both field measurements and from satellite, large scatter is likely observed. In addition, direct comparison was impossible because most field studies were carried out before the launch of the MODIS and SEVIRI instruments (van Leeuwen et al., 2014).

Given the large scale at which landscape fires occur and the high spatiotemporal variation in fuel consumption, satellite-derived fuel consumption estimates are crucial to get a better system understanding. While different satellite-derived fuel consumption estimates resulted in a similar spatial distribution, the absolute fuel consumption estimates remained more uncertain. This study clearly demonstrated the potential to derive fuel consumption estimates by combining satellite derive FRP and burned area. However, deriving accurate FRE estimates is difficult due to several sensor specific limitations. Here we choose to calibrate against field observations, correcting for such errors. Better understanding of for example the effect of tree cover on FRP detections would allow for expansion of such methods beyond open land cover types. Validation of the satellite-derived products by specifically designed field campaigns aiming for example at NPP or fire return period transects, or high resolution airborne remote sensing may further improve our understanding of the active fire sensor characteristics and provide more confidence in absolute fuel consumption estimates in the future.

## 5.2 Drivers and dynamics of fuel consumption

Fuel consumption depends on the amount of fuel available for burning and the combustion completeness. In arid areas available fuel and thus fuel consumption is often limited by precipitation. Across these arid and semi-arid areas precipitation generally determines vegetation productivity and tree cover. In more humid regions, however, fuel moisture may limit fuel consumption by lowering fire spread and the combustion completeness (Stott, 2000; van der Werf et al., 2008).




Consequently savannas of intermediate NPP and with a clear dry season tend to burn most frequently (Bowman et al., 2009). In our three study regions (South America, Sub-Saharan Africa and Australia) fires occurred under very different conditions in terms of NPP and fire return periods (Fig. 5a – c), partially as a result of the different distributions of NPP across the study regions. In South America most burned area occurred in regions with fire return periods between 3 – 8 years and intermediate productivity (800 – 1600 g DM $m^{-2}$ $yr^{-1}$). In Africa the vast majority of burned area was found in areas with short fire return periods (1 – 3 yrs) and a wide range of productivity (500 – 2000 g DM $m^{-2}$ $yr^{-1}$). And in Australia the majority of fires occurred in the more arid low productivity zones (<500 g DM $m^{-2}$ $yr^{-1}$) while annually burning regions were uncommon and restricted to the humid higher productivity zones (typical fire return periods were in the range 2 to 10 years). Although climate and vegetation shape the boundary conditions for fires to occur, most ignitions are of human origin (Scholes and Archer, 1997; Stott, 2000; Archibald et al., 2011; Scholes et al., 2011) and the differences between the continents are expected to be partly the result of different management practices. Overall, a pattern of increasing fuel consumption towards more productive regions and longer return periods was observed (Fig. 5d – f). Consequently fuel consumption in Africa, with short return periods, was relatively low compared to Australia or Southern America.

However, increases in fuel consumption with increasing time between fires and NPP were far from linear and other drivers also played a large role (e.g., Shea et al., 1996). Most notably, we found a clear difference between ecosystems where most fuel exists of grasses opposed to regions that were classified as shrubs. In Africa, the more arid regions with NPP below 500 g DM $m^{-2}$ $yr^{-1}$ are dominated by savannas or grasslands, while in Australia these regions are classified as shrubs (Fig. 5h and i). In the specific case of Australia, much of the interior is actually dominated by Hummock grasslands (rather than shrubs), grasses that functionally act like shrubs (Australian Native Vegetation Assessment, 2001) and are therefore classified as being shrublands in the UMD classification. Grasses may form well-connected fuel beds, resulting in short (often annual or biennial) fire return periods (Scholes and Archer, 1997; Beerling and Osborne, 2006; Archibald et al., 2013), while fire return periods in shrublands (or Hummock grasslands) were generally longer (Fig. 5). But on top of the differences in fire return periods between these low productivity ecosystems, the grass species that were dominant in most of Africa showed a rather slow fuel build up compared to shrubs or the Australian Hummock grasses even when fire return periods and productivity were similar (Fig. 5e and f). A possible explanation for the relatively slow fuel build up in African grasslands and savannas as opposed to Australian Hummock grasslands and shrublands could be grazing by livestock or wildlife and human management (Savadogo et al., 2007; Scholes et al., 2011). Shea et al. (1996) report a large impact of wildlife, ranging from insects to grazers, on fuel build up processes in various study sites in Africa and such effects will differ among continents given that neither South America or Australia have the diverse and dominant mega-herbivore fauna of Africa. Other differences may come from non-fire related decomposition rates, that depend on plant species and climate (Gupta and Singh, 1981).



In the more productive savannas marked differences were observed between the different continents (Fig. 5). Africa was unique when it comes to its short fire return periods, even in highly productive ecosystems. In African savannas of intermediate productivity (500 – 1500 g DM m$^{-2}$ yr$^{-1}$), fuel build up with time was slow compared to the other continents. These differences may originate from differences of grazing pressure or the occurrence of different species, as discussed above, but may also be related to management practices or climate. For example, the highest fuel consumption in the more humid African savannas was found in the most frequently burning grid cells, suggesting both high combustion completeness and high fuel availability. Short fire return periods provide a competitive advantage to herbaceous vegetation over woody vegetation (Bond et al., 2005; Bond, 2008). A high degree of canopy openness will result in more grass covered area and higher dry season ground surface temperatures and lower fuel moisture content resulting in high combustion completeness. However, a similar temperature driven effect may also be caused by the timing of the ignitions (Hoffa et al., 1999) directly related to management practices. Finally, the above assumes a stable situation of tree cover density and biomass over the study period, while in some regions there is tree cover loss due to increased fire frequency or land use change over our relatively short study period, while other areas are experiencing increases in tree cover (Wigley et al., 2010). A clear example was South America, where the apparent fuel build up (Fig. 5d) appears largely driven by high fuel consumption in active deforestation areas (Fig. 4; see Hansen et al., 2013). The effect of human management on fuel loads was also clearly visible in Africa's agricultural areas (e.g., Nigeria) where fuel loads were typically low (Fig. 4 and Fig. 5e and h).

Finally, the fuel consumption estimates derived here were compared to the modelled fuel consumption estimates of GFED4s. Within GFED fuel build up is largely driven by NPP and fire return periods, while biomass build up is distributed over two different pools: herbaceous and woody (van der Werf et al., 2010). This differentiation is important, because in savanna ecosystems most fires burn in the grass layer, leaving the older well established woody vegetation largely untouched (Scholes and Archer, 1997). Fuel consumption estimates derived here and by GFED were comparable in annual or biennial burning savannas (Fig. 6 and Fig. A1a - c). This is encouraging, because from an emissions perspective the modelling of fuel consumption has to be most accurate in areas that burn annually or biennial where little long-term fuel build up takes place. For arid areas in general but especially for shrublands and the Hummock grasslands in Australia, the fuel consumption estimates derived here were considerably larger than the ones estimated by GFED. Part of this difference may be caused by GFED using a universal fuel build up mechanism for all types of grasses and shrublands (van der Werf et al., 2010), which according to our findings seems oversimplified. In fact, Hummock grasses act like shrubs with bare soil between the mounds of hummock grass (Australian Native Vegetation Assessment, 2001), such behaviour likely results in very different fuel build up dynamics which may vary strongly depending on the wet season intensity as opposed to other grasses that form a well connected fuel bed. These results confirm the important role of arid and semi-arid drylands in the inter-annual variability of the global carbon cycle (Poulter et al., 2014).




In more humid regions, with higher woody cover, the fuel consumption estimates of GFED4s were higher than the ones derived here. Within GFED it is assumed that the amount of woody vegetation burned is a function of tree cover within savannas and woody savannas (van der Werf et al., 2010). It remains unclear to what extent the woody vegetation in savannas burns. Although fires greatly reduce the occurrence of trees in many savannas (Bond et al., 2005; Bond, 2008),

field studies often report that the established woody vegetation in savannas is rather resistant to fire (Scholes et al., 2011). The potential tree cover for a given area is directly related to mean annual precipitation (Sankaran et al., 2005), although further affected by e.g. the occurrence of different species (Lehmann et al., 2014) or availability of nutrients (Bond, 2008). In the tropics highest precipitation is generally found with decreasing dry season duration and may thus prevent fires from spreading to the woody fraction of the vegetation. Moreover, typical architecture of savanna trees varies considerably

between continents affecting their sensitivity to fire (Lehmann et al., 2014; Moncrieff et al., 2014). While some woody species may be better adjusted to relatively cool frequent fires with low fuel loads, most common in frequently burning and/or well grazed grasslands of Africa, other species are better adjusted to more intense and infrequent fire occurrence. Although fuel consumption estimates based on FRP detections may be affected by tree cover to some extent (Freeborn et al., 2014b), active deforestation areas in South America clearly stand out because of their high fuel consumption. We expect that

during future studies satellite-derived fuel consumption estimates may help to differentiate between grass fuelled fires and degradation fires, that additionally burn part of the woody cover. Moreover, satellite-derived fuel consumption estimates could be used as a reference for biogeochemical models, while providing improved insights in the underlying processes.

## 6 Conclusions

Satellite-derived fuel consumption estimates provide a unique opportunity to challenge current understanding of

spatiotemporal variation in fuel consumption that to date is mostly based on field studies and modelling. The fuel consumption estimates based on geostationary SEVIRI and polar orbiting MODIS fire radiative power (FRP) data showed good agreement in terms of spatial patterns, suggesting that these estimates were generally robust. When converting fire radiative energy (FRE) estimates derived from MODIS and SEVIRI to fuel consumption using a universal conversion factor based on laboratory measurements, fuel consumption estimates based on MODIS FRP data were about twice as high as the

ones based on the SEVIRI data. This can likely be attributed to SEVIRI failing to detect the more weakly burning, but highly numerous, smaller fires that ultimately are responsible for around half of the emitted FRE. On top of that, when we calibrated the FRE estimates based on MODIS FRP detections directly to field observations, we found a new conversion factor that was about 1.5 times larger than the one based on laboratory experiments. This discrepancy likely stems from underestimation of FRE based on the MODIS instruments, for example related to the decreased sensitivity of the instruments

towards the swath edges. Our best estimates of fuel consumption based on MODIS derived FRE using the conversion factor based on field observations were similar in magnitude as modelled fuel consumption estimates from GFED4s, but discrepancies were found in the spatial patterns. However, the limited number of field studies combined with the high




spatiotemporal heterogeneity of fuel consumption complicated the comparison of field studies with long-term coarse scale satellite-derived products, and uncertainty in absolute estimates remained therefore considerable. Field studies especially designed to validate satellite-derived fuel consumption estimates, aiming for example at NPP or fire return period transects, possibly using air-based remote sensing, could improve (confidence in) absolute fuel consumption estimates in the future.

Dominant biomass burning conditions in South America, Sub-Saharan Africa and Australia were highly different in terms of NPP and fire return periods, partly driving fuel consumption patterns. In South America most fires occurred in savannas with relatively long fire return periods, resulting in relatively high fuel consumption compared to the other study regions. In contrast, most burned area in Sub-Saharan Africa stemmed from (woody) savannas that burned annually or biennial with relatively low fuel consumption. Australian biomass burning was dominated by relatively unproductive (Hummock) grasslands with a wide range of fire return periods, while savannas with fire return periods of 2 – 3 years also contributed.

Besides NPP and fire return periods, vegetation type played an important role in determining the fuel build up mechanism. Grasslands favoured short fire return periods and were generally characterized by low fuel build up rates. Shrublands, or grassy species that functionally act like shrubs, on the other hand were generally characterized by longer return periods, but gradual fuel build up occurred over the years eventually leading to higher fuel consumption. Similarly, land management had a marked effect on fuel consumption. In the major deforestation regions of South America, fires consumed woody biomass during the MODIS era, increasing fuel consumption estimates. West African fuel consumption was clearly suppressed in some areas, likely associated with agriculture and/or grazing. These results demonstrate that the modelling of fuel consumption is complex while the relation between climate, vegetation and fuel consumption may vary across the continents depending on for example the presence of certain species. During future investigations satellite-derived fuel consumption estimates may be used as a reference dataset for biochemical models, and help to better understand the interaction between climate, vegetation patterns, landscape management and fuel consumption.

**Acknowledgements.** The authors would like to thank the data providing agencies (NASA and EUMETSAT LSA SAF) for making their data publicly available. This study was funded by the EU in the FP7 and H2020 projects MACC-II and MACC-III (contracts no. 283576 and 633080) and the European Research Council (ERC), grant number 280061.



## Appendix A

**Figure A1.** Spatial distribution of parameters affecting fuel consumption dynamics. (**a-c**) Fire return periods for South America, Sub-Saharan Africa and Australia, respectively; (**d-f**) net primary productivity for South America, Sub-Saharan Africa and Australia, respectively; and (**g-i**) dominant land cover type for South America, Sub-Saharan Africa and Australia, respectively.



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
