# Peer review of "Biomass burning fuel consumption dynamics in the tropics and subtropics assessed from satellite"

_Biogeosciences, 2015_

## Referee Comment (RC1) · M. Forkel (Referee) · 20 Jan 2016

Review of "Biomass burning fuel consumption dynamics in the (sub)tropics assessed from satellite" by Andela et al.

General comments

The paper presents an approach to estimate fuel consumption by combining satellite data with field data and analyzes the derived spatial patterns of fuel consumption. Burned area data from MODIS is combined with fire radiative power (FRP) data from SEVIRI and MODIS to estimate fuel consumption. This approach requires a factor to convert fire radiative energy (FRE) to burned dry matter. The authors used a standard factor reported in the literature based on laboratory measurements (e). As an alternative, the conversion factor was estimated from field data by fitting a regression between
[Figure]

MODIS FRE and field measurements of fuel consumption. By using the standard conversion factor, the derived conversion factor and by using MODIS or SEVIRI FRP data, the authors find similar spatial patterns of fuel consumption but large differences in absolute numbers. Relations between fuel consumption, NPP and fire return intervals remarkably differ between continents.

I very much appreciate the approach of combining these different datasets. Such estimates are certainly valuable to better understand and model vegetation-fire-carbon cycle interactions. The paper is very well written.

Specific comments

One conclusion of the authors is "Moreover, satellite-derived fuel consumption estimates could be used as a reference for biogeochemical models, while providing improved insights in the underlying processes." (p. 22, l. 16-17). Although I completely agree that satellite-derived data can help to improve process-representations in biogeochemical models, I disagree with this conclusion. The authors present large differences in fuel consumption between the MODIS- and SEVIRI-based estimates and additional large differences in the lab-based and the field-based FRE-to-DM conversion factor. These two issues indicate large uncertainties in fuel consumption estimates. Thus, I'm not convinced that these estimates can be used as a reference for models unless the uncertainties in fuel consumption are quantified. In my view, a major uncertainty originates from the fitted regression between MODIS FRE and field measurements of fuel consumption because only a limited set of field data is available with a limited representativeness for MODIS pixels. I think it is necessary to quantify the uncertainty of the regression (i.e. of the conversion factor) for example by bootstrapping the set of measurement point that goes into the computation of this regression. The bootstrapped distribution of conversion factors (or for example the 0.025, 0.5, and 0.975 quantiles of this distribution) can be then propagated into the computation of fuel consumption to provide spatial fields of upper and lower uncertainty estimates. The distribution of conversion factors can be also used to test if the lab-based factor

of 0.368 really differs from the derived factor or if this is a sampling issue. Such an uncertainty would make the estimate of fuel consumption much more valuable and I would accept it for benchmarking and testing biogeochemical models.

Minor remarks

p. 21, l. 31-32: I don't understand the connection of this sentence with the previous sentences. Can you please clarify it and improve the text.

- Figure 2 a, b and c: It seems that the two maps fit pretty well. I only noticed the biases after the second reading when I saw the labels of the color legend and the different axis ranges in (b). Can you please make the same color legend ranges for both maps and the same ranges for the axes in (b)?

- Figure 5 and corresponding analysis: Can you really treat fire return period as the independent variable? I assume fire return period and fuel consumption are highly inter-related. Maybe you can explain this better or you could use a different predictor variable. Additionally, it is strange that the high NPP values are at the bottom of the axis. The plot would be easier if NPP increases from bottom to top.

- Table 1: Can you add the references as additional column to improve readability?
* * *

---

## Referee Comment (RC2) · Anonymous Referee #2 · 17 Feb 2016

General Comments:

This paper describes an approach for combining satellite observations of active fires and burned areas to generate maps of average fuel load consumption (kg m-2) across South America, Africa, and Australia at 0.25 degree grid cell resolution. The ability to develop such a map is at the forefront of wildland fire science, and the production of an accurate map would certainly improve our understanding of global biomass burning and the pyrogenic carbon budget. However, as the authors demonstrate and fully admit, there is uncertainty in their map.

Uncertainties in estimates of fuel load consumption undoubtedly stem from the limitations of confidently measuring fire activity from satellite sensors. To address these satellite sensor limitations, the authors compare fuel load consumption estimates obtained from geostationary and polar orbiting platforms. However, as broached in the specific comments, it is not entirely apparent that their technique for aligning the active fire and burned area products completely restricts comparisons between the geostationary and polar orbiting observations to the same fire activity. Satellite sensor limitations are also addressed by calibrating satellite measurements with field measurements and deriving an alternative, sensor specific conversion factor relating radiant heat release and fuel consumption. However their intent for using an alternative conversion factor – rather than applying a bias correction to satellite-based estimates of FRP and FRE – deserves more consideration.

Aside from the techniques used to produce their map of fuel load consumption, the authors do not fairly acknowledge one aspect – and perhaps the one overarching aspect: the role that environmental conditions and fuel moisture contents play in fuel load consumption. Although NPP and time since fire are used to explore geographical differences in fuel load accumulation, little to no credit is given to the impact that environmental conditions and fuel moisture contents have on consumption completeness. It is very difficult to confidently interpret fuel load consumption estimates between geographical regions without knowing the environmental and fuel moisture conditions at the time of burning.

In my opinion, uncertainties in their fuel load consumption map due to satellite sensor limitations, and the somewhat incomplete interpretation of their map in the absence of fuel moisture contents, does not detract from the overall worthiness of this work. It is a very good start in the right direction, and at the very least, exposes areas for further refinement and opens arenas for further exploration. With some further clarification and explanation, I feel that this article can contribute to our understanding of global biomass burning and pave the way for more accurate fuel load consumption maps in the future.

Specific Comments:

1. Page 2, Line 19: For brevity the authors refer to fuel consumption per unit area burned (kgm-2) simply as fuel consumption. Granted the authors state this up front, but this is the only place that it is mentioned. Anyone skipping the introduction and skimming the methods and results might confuse the traditional sense of fuel consumption (kg) with the authors definition of fuel consumption (kgm-2). Moreover, in the conclusions on Page 22 Lines 22-25, are the authors talking about fuel consumption (kg) or fuel consumption (kgm-2)? The pre-burn mass of fuel per unit area (kgm-2) is typically referred to as the "fuel load". The authors should consider whether or not the term "fuel load consumption" better describes what they are attempting to estimate. In my opinion, the terms "fuel load consumption" and "fuel load consumed" are more accurate descriptors that pose less of a chance for confusion.

2. Page 4, Line 26 – Page 5, Line 2: Does a description of the "mean fire return period" belong in the Data section, or should it be moved to the Methods section? Also, I understand that the authors are trying to quantify the amount of time between fires as a way of explaining fuel accumulation and eventually fuel load reduction. However I think they may have their terms confused, and I'm not entirely clear on how the "mean fire return period" is calculated. The authors state that: "We estimated the mean fire return period based on the 14 years of MCD64A1 burned area data, by recording how many times each 500 m resolution MODIS grid cell had burned during the 2001 – 2014 period and then dividing this by the 14 years." According to this definition, the "mean fire return period" in a 500 m grid cell can range 0.07 "fires" per year if the grid cell burned once to 1.00 "fire" per year if the grid cell burned every year (with 0's excluded). First things first: metrics with units of inverse time are frequencies. Rather than calculating the "fire return period," it seems to me that the authors are calculating the fire frequency of a 500 m grid cell. The inverse of the fire frequency is the fire return period, which in this case would range from 1 yr if the grid cell burned every year to 14 yrs if the grid cell burned only once during the study period. On Page 12, Lines 23-24, the authors report a "mean fire return period 1.75 years", leading me to believe that they are computing a frequency, but reporting a period. Please confirm?

Also, the authors state that: "We then calculated the mean fire return period for each 0.25° grid cell as the mean return period of all 500 m grid cells within each 0.25° grid cell, weighted by burned area." Again, from their description, I would expect the "mean fire return period" to be considerably less than 1.00 unless every 500 m pixel within a 0.25 grid cell burned every year, which doesn't agree with values such as 3 – 8 years reported on Page 15, Lines 13-19. Beyond the period/frequency issue, I do not understand how the "mean fire return period" in a 0.25 degree grid cell is weighted by burned area. Please elaborate. To me, it almost sounds like the authors are trying to calculate a "fire rotation", or the amount of time it takes to burn an area equal to the size of the study area. All in all, the authors should confirm and clarify their computation of a "mean fire return period" for a 0.25 degree grid cell.

3. Page 7, Lines 11-20: It seems to me that aligning the active fire and burned area products is absolutely crucial to the estimation of fuel load reduction. Although aligning the active fire and burned area pixels is technically feasible, the underlying "squishy" part of this process (which the authors briefly touch upon) is ensuring that the FRP detected by SEVIRI is only emitted from burned areas detected by MODIS, and conversely, that all the burned areas detected by MODIS contribute to some of the FRP measured by SEVIRI. Can the authors perform a sensitivity analysis to quantify the impact of the 15- day window on their estimates of fuel load consumption? It seems to me that expanding the 15-day window around the burned area detection date would result in more active fire pixels associated with the same burned area and thus result in higher estimates of fuel load consumption. Similarly, contracting the 15-day window would result in lower fuel load consumption estimates. Please confirm, and consider warning the reader about the sensitivity of fuel load consumption estimates to the 15-day window. Also, SEVIRI grid cells with burned area detections but no active fire detections were excluded from the analysis. However the authors never describe how they treat SEVIRI active fire pixels with no corresponding MODIS burned area detections. Were there any?

4. Page 7, Line 23: Please confirm the value of the conversion factor. There are several instances that reference a value of 0.356 kg MJ-1, and there are several other instances that reference a value of 0.368 kg MJ-1. Which value are you using?

5. Page 7, Lines 22-29: Please see my comments concerning the interpretation of the "mean fuel load consumption" calculated using observations accumulated over long time periods (Page 8, Line 27 – Page 9, Line 2).

6. Page 8, Line 27 – Page 9, Line 2: Yes, I agree with the authors here. However I think they are overlooking a critically important aspect. Accumulating observations over long time periods (e.g., over many years) precludes a seasonal analysis. For the moment, consider a hypothetically static pre-burn fuel load that does not vary from the end of one rainy season to the beginning of the next rainy season. For a constant pre-burn fuel load that does not change over time, fuel load consumption will still vary depending on when during the dry season the landscape burns due to seasonal oscillations in fuel moisture contents, which drive seasonal oscillations in consumption completeness (Hoffa et al., 1999). Accumulating observations over long time frames fails to resolve the seasonal oscillations in consumption completeness and thus fuel load consumption. I strongly suggest that the authors warn the reader that estimates of fuel load consumption calculated from observations accumulated over long time periods are more representative of values observed at the peak in fire activity when the satellites detect the most active fire pixels and burned area pixels within a 0.25 degree grid cell.

Here's the really important bit though: the seasonality of fire activity is not always synchronized with the seasonality of fuel moisture contents and consumption completeness (Le Page et al., 2010). Hence peaks in fire activity may not always coincide with identical fuel moisture conditions. Across the majority of Brazil, for example, the peak in fire activity generally occurs when fuels are driest (i.e., the middle of the fire season coincides with the middle of the CBI season, according to Figure 4 of Le Page et al., 2010). In contrast, across much of Africa, the middle of the fire season occurs before

the middle of the CBI season. Therefore, even if the pre-burn fuel loads are identical between South America and Africa, the consumption completeness (%) at the time of peak fire activity would differ between the two locations due to differences in the seasonal synchronization of fire activity and fire weather, which would then lead to different estimates of fuel load consumption.

The authors do a nice job of using NPP and time since fire to explain geographical differences in the pre-burn fuel load, however they do not account for differences in consumption completeness (%), a value just as important in traditional estimates of fuel load consumption. In my opinion, the inability to identify fuel moisture conditions at the time of burning hinders a complete and confident interpretation of the geographical differences in fuel load consumption. Aligning the active fire and burned area pixels with a map of fuel moistures at the time of burning would be the ideal solution. However if the authors forgo such an analysis, they should at least make it extremely clear to the reader that fuel load consumption depends on the pre-burn fuel load AND consumption completeness, and that the latter is influenced by the environmental conditions and fuel moisture contents at the time of burning, which are not accounted for here.

Hoffa, E. A., Ward, D. E., Hao, W. M., Susott, R. A. and Wakimoto, R. H.: Seasonality of carbon emissions from biomass burning in a Zambian savanna, J. Geophys. Res., 104, 13841 – 13853, doi:10.1029/1999JD900091, 1999.

Le Page, Y., Oom, D., Silva, J. M. N., Jonsson, P., and Pereira, J. M. C.: Seasonality of vegetation fires as modified by human action: observing the deviation from eco-climatic fire regimes Global Ecol. Biogeogr., 19, 575–588, 2010.

7. Page 10, Line 9 – Page 11, Line 14: Since the MCD64A1 product is used for both estimates, can it be assumed that the differences between FC MODIS and FC SEVIRI are entirely attributed to the different active fire products and the different methods for converting FRP to FRE?

8. Page 10, Line 9 – Page 11, Line 14: I'm curious about how much overlap there is in

the fire activity that's driving the two estimates of fuel load consumption. I mean, are FC MODIS and FC SEVIRI driven by the same fire activity, or are they two different sets of fire activity, or is the fire activity that that drives FC SEVIRI a subset of the fire activity that drives FC MODIS? Based on the author's statement on Page 8, Line 21: "In contrast to the approach based on SEVIRI data, here all burned area observations were included", it would seem to me that FC SEVIRI is driven by fire activity that is a subset of the fire activity that is driving FC MODIS, if FC MODIS is limited to 2010-2014. Can any insights be gained by limiting the calculation of FC MODIS to the time period used to calculate FC SEVIRI (2010 – 2014)? Perhaps not the bias, but it seems to me that the scatter in the relationship between FC MODIS and FC SEVIRI could be due to the possibility that MODIS and SEVIRI are observing different fire activity within the same 0.25 grid cell. I may be mistaken, but I don't think this was ever offered as an explanation for the scatter. Unless the authors can demonstrate that estimates of FC MODIS and FC SEVIRI are driven by the same fire activity, then I think they have to concede that the scatter in the relationship could be attributed to the possibility that MODIS and SEVIRI are observing different fires within the same grid cell. Note that if fuel load consumption is homogeneous within a 0.25 degree grid cell, then it doesn't matter what fire activity MODIS and SEVIRI observed. However by their own admission, fuel load consumption is heterogeneous, and therefore has the potential to induce scatter in the relationship between FC MODIS and FC SEVIRI if MODIS and SEVIRI observe different fires within the same grid cell.

9. Page 12, Lines 7 – 14: I think this section raises a very interesting question: is it reasonable to derive alternative conversion factors (kgMJ-1) for different satellite sensors? To be honest, I don't know the answer to this question, so I leave it to the authors to pontificate. Note that the authors specifically state that: "Although the FRE per unit area burned can be converted to fuel consumption using the conversion factor found by Wooster et al. (2005) during laboratory experiments which we have used so far, some instrument specific issues may further affect the FRE estimates from space (see methods). In order to correct for uncertainties in the MODIS derived FRE estimates,

we derived an alternative conversion factor by comparing the MODIS FRE per unit area burned directly to field measurements instead (Fig. 3a)." Here the authors admit that the reason for deriving an alternative conversion factor is because of the biases and uncertainties of estimating FRE from MODIS. If this is the case, then shouldn't the adjustment be more appropriately applied to the detection and conversion of FRP to FRE? Why should it be necessary to adjust an empirically derived "physical constant" due to satellite and sensor limitations? Are the authors suggesting that satellite sensor artefacts be incorporated into combustion chemistry and the relationship between radiant heat release and fuel consumption? Note that 0.356 kg MJ-1 (or 0.368 kgMJ-1) is a laboratory derived value, and as such its inverse represents (as close as possible) the amount of radiant heat released per kg of fuel consumed. Adjusting this value to 0.572 kg MJ-1 means that its inverse has a different interpretation: the amount of radiant heat measured by MODIS per kg of fuel consumed. The difference in the meaning is subtle, but not trivial. For instance, there's a difference between (a) the radiant fraction, and (b) the fraction of total heat released that is measured as radiation by MODIS. The other option is to use the laboratory relationship between radiant heat and fuel consumption (which has a universal, physical meaning), and separately derive a MODIS specific FRP-FRE-adjustment factor to account for sensor limitations. I'll leave it to the authors to explain why deriving alternative, sensor specific conversion factors– as performed here – is a better option than using field measurements to derive an adjustment factor that's applied during the conversion of FRP to FRE. Although the two calibration options will obviously give you the same results, they nevertheless have different meanings.

10. Page 15, Lines 8 – 20. To follow on from my previous comment, the authors recognize the role of productivity and fire return periods on the accumulation of fuels, and thus account for the impact of the pre-burn fuel load on fuel load consumption. However, the authors do not equally acknowledge the role of environmental and fuel moisture conditions at the time of burning and their influence on consumption completeness. Ideally the analysis should account for fuel moisture contents at the time of

burning. If such an analysis is not feasible, then the authors should make every effort to remind the reader of the impact of different fuel moisture conditions on consumption completeness and thus fuel load reduction.

11. Page 18, Lines 15-17: Couldn't the "large natural temporal variation in fuel consumption combined with the different periods of data availability" be discounted if FC MODIS and FC SEVIRI are compared between 2010-2014? Also, in addition to the different sensor characteristics, aren't there differences in the methods for converting MODIS and SEVIRI measurements of FRP to FRE (i.e., dividing by detection opportunities vs. temporal integration)?

12. Page 19, Line 30 – 34: This is one of only a few places where the authors disclose to the reader that fuel load consumption depends on the pre-burn fuel load and consumption completeness.

13. Page 20, Line 9 – 15: Indeed. Due to different management goals, fires are lit earlier in the dry season in Africa compared to South America, and particularly in Brazil where fires are generally lit at the peak of the dry season (as shown in Figure 4 of Le page et al., 2010). Hence fire management practices not only determine the fire return period, which affects the fuel load accumulation, but fire management practices also determine what time of year the fires are lit, and thus under what fuel moisture conditions the fires burn. Whilst the authors conclude that fuel load consumption across Africa is relatively low compared to Australia or South America due to differences in the fire return periods, one could also argue that based on the maps presented by Le Page et al. (2010), fuel load consumption is also lower in Africa since fires are more often lit earlier in the dry season when fuel moistures are higher and consumption completeness is lower.

Technical Corrections:

1. Page 1, Line 23-24: Incomplete sentence or incomplete thought.

2. Page 2, Line 19-21: Consider changing the sentence to read: "...is a key indicator of the consequences of changing management practices, vegetation characteristics and climate on fire regimes, as well as a key parameter required in fire emissions estimates."

3. Page 2, Line 21-23: Consider changing the sentence to read: "Yet, spatiotemporal dynamics of fuel consumption on a continental scale remain largely unmeasured and poorly understood (van Leeuwen et al., 2014)."

4. Page 3, Line 23: Consider changing the sentence to read: "...creating the first fully satellite-derived fuel consumption map for Africa."

5. Page 4, Line 1-2: Consider changing the sentence to read: "...in an attempt to provide more statistically representative fuel consumption estimates, particularly in less frequently burned grid cells.

6. Page 4, Line 7-8: Consider changing the sentence to read: "Finally, we used our fuel load reduction map to explore the drivers of fuel consumption in the study regions."

7. Page 4, Line 19-24: It may be helpful to remind the reader here that the MCD64A1 burned area product is also used by GFED.

8. Page 6, Line 24: Consider changing the sentence to read: "...first derived a fuel consumption map for Sub-Saharan Africa."

9. Page 8, Line 33: Grammar, pluralize: "Minimising the impact of these types of perturbations..."

---

## Author Comment (AC1) · 28 Apr 2016

Review of "Biomass burning fuel consumption dynamics in the (sub)tropics assessed from satellite" by Andela et al.

**General comments**

The paper presents an approach to estimate fuel consumption by combining satellite data with field data and analyzes the derived spatial patterns of fuel consumption. Burned area data from MODIS is combined with fire radiative power (FRP) data from SEVIRI and MODIS to estimate fuel consumption. This approach requires a factor to convert fire radiative energy (FRE) to burned dry matter. The authors used a standard factor reported in the literature based on laboratory measurements (e). As an alternative, the conversion factor was estimated from field data by fitting a regression between MODIS FRE and field measurements of fuel consumption. By using the standard conversion factor, the derived conversion factor and by using MODIS or SEVIRI FRP data, the authors find similar spatial patterns of fuel consumption but large differences in absolute numbers. Relations between fuel consumption, NPP and fire return intervals remarkably differ between continents.

I very much appreciate the approach of combining these different datasets. Such estimates are certainly valuable to better understand and model vegetation-fire-carbon cycle interactions. The paper is very well written.

*We would like to start thanking the reviewer for his thoughtful and constructive remarks. Please see our response to the specific comments and minor remarks below. In addition, we will upload a pdf file containing the suggested textual changes in response to both reviews (using track change) and the updated figures, references to line numbers used here refer to this document.*

**Specific comments**

One conclusion of the authors is "Moreover, satellite-derived fuel consumption estimates could be used as a reference for biogeochemical models, while providing improved insights in the underlying processes." (p. 22, l. 16-17). Although I completely agree that satellite-derived data can help to improve process-representations in bio-geochemical models, I disagree with this conclusion. The authors present large differences in fuel consumption between the MODIS- and SEVIRI-based estimates and additional large differences in the lab-based and the field-based FRE-to-DM conversion factor. These two issues indicate large uncertainties in fuel consumption estimates.

Thus, I'm not convinced that these estimates can be used as a reference for models unless the uncertainties in fuel consumption are quantified. In my view, a major uncertainty originates from the fitted regression between MODIS FRE and field measurements of fuel consumption because only a limited set of field data is available with a limited representativeness for MODIS pixels. I think it is necessary to quantify the uncertainty of the regression (i.e. of the conversion factor) for example by bootstrapping the set of measurement point that goes into the computation of this regression. The bootstrapped distribution of conversion factors (or for example the 0.025, 0.5, and 0.975 quantiles of this distribution) can be then propagated into the computation of fuel consumption to provide spatial fields of upper and lower uncertainty estimates.

The distribution of conversion factors can be also used to test if the lab-based factor of 0.368 really differs from the derived factor or if this is a sampling issue. Such an uncertainty would make the

estimate of fuel consumption much more valuable and I would accept it for benchmarking and testing biogeochemical models.

*We agree that the uncertainty in the absolute fuel consumption estimates remains high. Our confidence in the spatial patterns is better because we find a reasonable comparison between the spatial patterns of SEVIRI-derived fuel consumption (observing the full diurnal cycle) and MODIS derived fuel consumption. The biggest challenge is to create accurate estimates of FRE, while SEVIRI misses much fire activity due to its large distance to Earth and subsequently large pixel size, the MODIS instruments provide only a limited number of daily observations and further suffer from increasing pixel sizes towards the swath edges. We assume that the conversion factor found by Wooster et al. (2005) is correct, and that structural differences between the uncorrected MODIS derived fuel consumption and field observations stem from errors in our FRE estimates. Following the suggestion of reviewer #2 we therefore now speak of a "FRE correction factor", rather than deriving an alternative conversion factor. Although this does not affect our fuel consumption estimates, it does provide the reader with better insight in where the largest uncertainties originate from.*

*We appreciate the suggestion to use bootstrapping to get an estimate of the uncertainty associated with the "FRE correction factor". We now show the 95% confidence interval of the FRE correction factor in Fig. 3a and we have also made updates to our methods, results and discussion sections accordingly. However, we expect that this may be a relative conservative estimate of the total uncertainty, for example because of the difficulties associated with the comparison of field observations with our long term mean 0.25° estimate and the fact that uncertainty may further be affected by the fire diurnal cycle, regionally. We found the "FRE correction factor" to be 1.56, indicating that MODIS derived FRE per unit area burned should be multiplied by a factor of 1.56 to get fuel consumption of the same magnitude as the corresponding field studies. We expect that the decreasing sensitivity of the MODIS instruments towards the swath edges (e.g., Freeborn et al., 2011) is responsible for most of the underestimation of MODIS-derived FRE. Using bootstrapping (n=10,000, method="bias corrected and accelerated bootstrap") we found that the 95% confidence interval of the bootstrapped distribution of the slopes is 1.30 - 1.80. This corresponds to a 16% increase, or decrease, in absolute fuel consumption estimates.*

**Minor remarks**

**p. 21, l. 31-32:** I don't understand the connection of this sentence with the previous sentences. Can you please clarify it and improve the text.

*We mean to say that compared to GFED we find relatively high fuel consumption in many of the more arid regions. This finding suggests that these regions may play a more important role in the inter annual variability of the land carbon sink than what would be expected based on current state of the art biogeochemical models, like GFED4. We have changed the sentence to read:*
*"The enhanced fuel consumption in arid and semi-arid drylands found here confirms the important role of arid and semi-arid drylands in the inter-annual variability of the global carbon cycle (Poulter et al., 2014)."*

**Figure 2 a, b and c:** It seems that the two maps fit pretty well. I only noticed the biases after the second reading when I saw the labels of the color legend and the different axis ranges in (b). Can you please make the same color legend ranges for both maps and the same ranges for the axes in (b)?

*The goal of this figure is to show the reader that the spatial patterns in fuel consumption derived from SEVIRI and MODIS are rather similar, despite the large absolute differences. We discuss the absolute differences both in the results (P11, L11-12) and in the discussion (P20, L19-22). To avoid further confusion we now specifically mention this difference in the caption of Fig. 2: "Note that on average MODIS-derived FC is about twice as large as SEVIRI-derived FC."*

**Figure 5 and corresponding analysis:** Can you really treat fire return period as the independent variable? I assume fire return period and fuel consumption are highly inter-related. Maybe you can explain this better or you could use a different predictor variable. Additionally, it is strange that the high NPP values are at the bottom of the axis. The plot would be easier if NPP increases from bottom to top.

*We have chosen NPP and fire return periods because they are often thought of as the main drivers of fuel loads and are responsible for much of the modelled variation in fuel consumption in GFED4s. In more arid areas combustion completeness is often high and fire return period can most likely be thought of as an independent explanatory variable. In more humid regions with more incomplete combustion an effect of fuel consumption on fire return periods may be expected. For example, when lower fuel loads are associated with higher flammability and thus likelihood of burning. We have now updated the text in the discussion to better acknowledge the possible interaction between fuel consumption, fuel loads and fire return periods.*

*P23, L19-24 "For example, the highest fuel consumption in the more humid African savannas was found in the most frequently burning grid cells, suggesting a high combustion completeness. In areas where burning is largely limited by fuel humidity, the combustion completeness may have a considerable impact on fuel consumption. The fact that both frequently burning and almost fire free areas occur under similar climatic conditions in (sub)tropical savannas suggest that fuel conditions are important, while frequent fire occurrence may enhance flammability (Shea et al., 1996; Ward et al., 1996)."*

*In addition, we have also reversed the y-axis of Figure 5.*

**Table 1:** Can you add the references as additional column to improve readability?
*- Done*

**References**

Freeborn, P. H., Wooster, M. J. and Roberts, G.: Addressing the spatiotemporal sampling design of MODIS to provide estimates of the fire radiative energy emitted from Africa, Remote Sens. Environ., 115, 475–489, 2011.

Poulter, B., Frank, D., Ciais, P., Myneni, R. B., Andela, N., Bi, J., Broquet, G., Canadell, J. G., Chevallier, F., Liu, Y. Y., Running, S. W., Sitch, S. and van der Werf, G. R.: Contribution of semi-arid ecosystems to interannual variability of the global carbon cycle, Nature, 509, 600–603, 2014.

Shea, R. W., Shea, B. W., Kauffman, J. B., Ward, D. E., Haskins, C. I. and Scholes, M. C.: Fuel biomass and combustion factors associated with fires in savanna ecosystems of South Africa and Zambia, J. Geophys. Res., 101, 23551 – 23568, 1996.

Ward, D. E., Hao, W. M., Susott, R. A., Babbitt, R. E., Shea, R. W., Kauffman, J. B. and Justice, C. O.: Effect of fuel composition on combustion efficiency and emission factors for African savanna ecosystems, J. Geophys. Res., 101, 23569 – 23576, 1996.

---

## Author Comment (AC2) · 28 Apr 2016

**General Comments:**

This paper describes an approach for combining satellite observations of active fires and burned areas to generate maps of average fuel load consumption (kg m-2) across South America, Africa, and Australia at 0.25 degree grid cell resolution. The ability to develop such a map is at the forefront of wildland fire science, and the production of an accurate map would certainly improve our understanding of global biomass burning and the pyrogenic carbon budget. However, as the authors demonstrate and fully admit, there is uncertainty in their map.

Uncertainties in estimates of fuel load consumption undoubtedly stem from the limitations of confidently measuring fire activity from satellite sensors. To address these satellite sensor limitations, the authors compare fuel load consumption estimates obtained from geostationary and polar orbiting platforms. However, as broached in the specific comments, it is not entirely apparent that their technique for aligning the active fire and burned area products completely restricts comparisons between the geostationary and polar orbiting observations to the same fire activity. Satellite sensor limitations are also addressed by calibrating satellite measurements with field measurements and deriving an alternative, sensor specific conversion factor relating radiant heat release and fuel consumption. However their intent for using an alternative con-version factor – rather than applying a bias correction to satellite-based estimates of FRP and FRE – deserves more consideration.

Aside from the techniques used to produce their map of fuel load consumption, the authors do not fairly acknowledge one aspect – and perhaps the one overarching aspect: the role that environmental conditions and fuel moisture contents play in fuel load consumption. Although NPP and time since fire are used to explore geographical differences in fuel load accumulation, little to no credit is given to the impact that environmental conditions and fuel moisture contents have on consumption completeness. It is very difficult to confidently interpret fuel load consumption estimates between geographical regions without knowing the environmental and fuel moisture conditions at the time of burning.

In my opinion, uncertainties in their fuel load consumption map due to satellite sensor limitations, and the somewhat incomplete interpretation of their map in the absence of fuel moisture contents, does not detract from the overall worthiness of this work. It is a very good start in the right direction, and at the very least, exposes areas for further refinement and opens arenas for further exploration. With some further clarification and explanation, I feel that this article can contribute to our understanding of global biomass burning and pave the way for more accurate fuel load consumption maps in the future.

*We would like to start thanking the reviewer for his thoughtful and constructive remarks. Please see our response to the specific comments and technical corrections below. In addition, we will upload a pdf file containing the suggested textual changes in response to both reviews (using track change) and the updated figures, references to line numbers used here refer to this document.*

**Specific Comments:**

**1. Page 2, Line 19:**

For brevity the authors refer to fuel consumption per unit area burned (kgm-2) simply as fuel consumption. Granted the authors state this up front, but this is the only place that it is mentioned. Anyone skipping the introduction and skimming the methods and results might confuse the traditional sense of fuel consumption (kg) with the authors definition of fuel consumption (kg m-2). Moreover, in the conclusions on Page 22 Lines 22-25, are the authors talking about fuel consumption (kg) or fuel consumption (kgm-2)? The pre-burn mass of fuel per unit area (kgm-2) is typically referred to as the "fuel load". The authors should consider whether or not the term "fuel load consumption" better describes what they are attempting to estimate. In my opinion, the terms "fuel load consumption" and "fuel load consumed" are more accurate descriptors that pose less of a chance for confusion.

*To our knowledge "Fuel consumption" is generally defined as:*
*"A direct measurement of how much biomass was consumed or volatilized in a fire, usually expressed as a mass per unit area (on a dry basis). Biomass loading x combustion factor is one way to estimate fuel consumption. Another way is to average the relevant fuel consumption measurements."*
*See "A Consistent Set of Biomass Burning Terminology" by Yokelson et al., but also the literature cited in our BGD paper.*

*In our manuscript all instances of 'fuel consumption' have the same physical meaning and units of kg $m^{-2}$. However, we appreciate that differences in the interpretation of such concepts may exist. We now repeat our definition of "fuel consumption" at the beginning of the discussion and conclusions sections, in addition the units are mentioned on all the figures and in the table.*

**2. Page 4, Line 26 – Page 5, Line 2:**

Does a description of the "mean fire return period" belong in the Data section, or should it be moved to the Methods section? Also, I understand that the authors are trying to quantify the amount of time between fires as a way of explaining fuel accumulation and eventually fuel load reduction. However I think they may have their terms confused, and I'm not entirely clear on how the "mean fire return period" is calculated. The authors state that: "We estimated the mean fire return period based on the 14 years of MCD64A1 burned area data, by recording how many times each 500 m resolution MODIS grid cell had burned during the 2001 – 2014 period and then dividing this by the 14 years." According to this definition, the "mean fire return period" in a 500 m grid cell can range 0.07 "fires" per year if the grid cell burned once to 1.00 "fire" per year if the grid cell burned every year (with 0's excluded). First things first: metrics with units of inverse time are frequencies. Rather than calculating the "fire return period," it seems to me that the authors are calculating the fire frequency of a 500 m grid cell. The inverse of the fire frequency is the fire return period, which in this case would range from 1 yr if the grid cell burned every year to 14 yrs if the grid cell burned only once during the study period. On Page 12, Lines 23-24, the authors report a "mean fire return period 1.75 years", leading me to believe that they are computing a frequency, but reporting a period. Please confirm?

Also, the authors state that: "We then calculated the mean fire return period for each 0.25° grid cell as the mean return period of all 500 m grid cells within each 0.25° grid cell, weighted by burned area." Again, from their description, I would expect the "mean fire return period" to be considerably less than 1.00 unless every 500 m pixel within a 0.25° grid cell burned every year, which doesn't agree with values such as 3 – 8 years reported on Page 15, Lines 13-19. Beyond the period/frequency issue, I do not understand how the "mean fire return period" in a 0.25 degree grid cell is weighted by burned area. Please elaborate. To me, it almost sounds like the authors are trying to calculate a "fire rotation", or the amount of time it takes to burn an area equal to the size of the study area. All in all, the authors should confirm and clarify their computation of a "mean fire return period" for a 0.25 degree grid cell.

*We apologize for our confusing description of how we estimate the fire return period. In contrast to what is stated in the discussion paper (".. and then dividing this by 14 years.") we have estimated the fire return period for each 500 m pixel by dividing 14 by the number of times that the pixel burned (units are thus yr and not yr$^{-1}$). A fire return period of 1.75 years thus correspond to a 500 m pixel that has burned 8 times during the 14 year study period.*
*Considering the estimation of the mean fire return period per 0.25° grid cell, what we mean to say by "weighted by burned area" is that a 500 m pixel that burned every year (fire return period of 1 yr) causes 14 times the burned area of a pixel that only burned once during the study period (fire return period of 14 yrs). So the "mean fire return period weighted by burned area" of these two pixels would be (1\*14 + 14\*1)/15 = 1.86 years. Although this way the mean fire return period will be more heavily influenced by the frequently burning grid cells, the mean fuel consumption estimate is calculated in a similar manner, and we therefore expect that this way the fire return period may better explain the spatial distribution of our fuel consumption estimates.*

*We made the following textual changes, to provide a better explanation of how we estimate the fire return period at 0.25° spatial resolution:*
*Page 4, line 31 – page 5, line 1 "The fire return period is estimated as 14 divided by the number of times that a given grid cell burned during the 14 year study period."*
*Page 5, lines 5 – 10 "We then calculated the mean fire return period for each 0.25° grid cell as the mean return period of all 500 m grid cells within each 0.25° grid cell, weighted by burned area. When estimating the mean fire return period per 0.25° grid cell, a 500 m grid cell with a fire return period of 1 year (burning 14 times during the study period) was thus assigned a weight 14 times larger than a 500 m grid cell that burned only once during the study period (having a fire return period of 14 years). We decided to weigh the fire return period by burned area to facilitate the interpretation of the mean fuel consumption conditions, that will in a similar way be dominated by the most frequently burning grid cells."*

**3. Page 7, Lines 11-20:**
It seems to me that aligning the active fire and burned area products is absolutely crucial to the estimation of fuel load reduction. Although aligning the active fire and burned area pixels is technically feasible, the underlying "squishy" part of this process (which the authors briefly touch upon) is ensuring that the FRP detected by SEVIRI is only emitted from burned areas detected by MODIS, and conversely, that all the burned areas detected by MODIS contribute to some of the FRP measured by SEVIRI. Can the authors perform a sensitivity analysis to quantify the impact of the 15-

day window on their estimates of fuel load consumption? It seems to me that expanding the 15-day window around the burned area detection date would result in more active fire pixels associated with the same burned area and thus result in higher estimates of fuel load consumption. Similarly, contracting the 15-day window would result in lower fuel load consumption estimates. Please confirm, and consider warning the reader about the sensitivity of fuel load consumption estimates to the 15-day window. Also, SEVIRI grid cells with burned area detections but no active fire detections were excluded from the analysis. However the authors never describe how they treat SEVIRI active fire pixels with no corresponding MODIS burned area detections. Were there any?

*We agree that the 15 day window is somewhat arbitrary and window size will affect the amount of FRE associated with a certain area burned. The relatively large window (15 days) was chosen because of the possible uncertainty in the burn date (see Fig. 11 in Giglio et al., 2013), and because a fire may burn for several days before the full grid cell is burned. The latter issue may be more important in case of the SEVIRI data, with a spatial resolution of 3x3 km at nadir, than for MODIS with a spatial resolution of 1x1 km. Roberts et al. (2011) investigated this issue and show the distribution of the active fire detections around the estimated day of burn by the burned area product (see Fig. 1 in Roberts et al., 2011). They find that >80% of SEVIRI active fire detections occur within 2 days before or after the day of burn as determined by the burned area product, after which the sensitivity rapidly decreases. As may be expected the curve was positively skewed, and more than 50% of the active fire detections occur after the burned area detection. The effect of increasing the time window of 15 days as used in our BGD paper would thus be small, the effect of a small decrease in the time window would also be limited but it is crucial to maintain an absolute minimum of 5 days.*

*In addition it is possible that two active fires burn at the same time within a given >9 km² SEVIRI grid cell. If one of the two fires leaves burned area and the other does not this could in theory cause an overestimation of FRE per unit of area burned and thus of fuel consumption. Given Fig. 1 of Roberts et al. (2011) we argue that this effect is likely small because of: (i) the shape of the curve and the restricted time window, and (ii) the fire that leaves detectable burned area will in general be significantly larger than the fire that does not and will therefore be responsible for most of the FRE. Also, it may occur that burned area is observed by MODIS but no corresponding FRE by the SEVIRI instrument. This was the case for ~3% of annual burned area and this data was excluded because it was impossible to estimate fuel consumption if no FRP was observed. We expect that these instances were mostly related to periods of cloud cover or fires burning at low intensity. The fact that only 3% of annual burned area had no corresponding FRE, while a much larger fraction of the observed FRE (by SEVIRI) had no corresponding burned area suggests that although the sensitivity of the SEVIRI instrument to small fires is much smaller than that of the MODIS instruments, it may still be larger than that of the burned area product. Therefore we assume that instances of two fires burning within the 15 day time window and a given ~9 km² SEVIRI grid cell that both leave burned area but of which only one is detected by the SEVIRI instrument are rare. Nonetheless it is clearly possible, especially under conditions of cloud cover. Although it is an interesting and important discussion, we expect that other sensor specific aspects (e.g., related to the pixel size and the ability to detect low FRP fires) are the main source of errors.*

*In the updated manuscript we discuss the possible effects of the 15-day time window and of excluding burned area without corresponding FRP in more detail:*

*Page 7, lines 23-33 "Because of the uncertainty of the burn date in the burned area product (Boschetti et al., 2010; Giglio et al., 2013), and the fact that a fire can burn multiple days, we followed Roberts et al. (2011) and assumed that all FRP detections within one week of the burned area observations (before or after; i.e., in total 15 days) in a given grid cell belonged to the same fire. Roberts et al. (2011) investigated the distribution of active fire detections in the days around the "day of burn" as determined by the burned area data set and showed that >80% of the SEVIRI active fire detections occurred within 2 days before or after the day of burn, after which the sensitivity rapidly decreases. Using a 15 day time window thus includes nearly all FRE that can be associated with a given fire, while the possible effect of small fires with observed FRP but without corresponding burned area (burning within the same pixel and time window) on the fuel consumption estimates is likely small. Grid cells having only burned area observations but no corresponding FRP detections are likely related to fires having relatively low FRP or those that were obscured by clouds (Roberts et al., 2011). These areas (3% of annual burned area) were excluded from our analysis."*

**4. Page 7, Line 23:**
Please confirm the value of the conversion factor. There are several instances that reference a value of 0.356 kg MJ-1, and there are several other instances that reference a value of 0.368 kg MJ-1. Which value are you using?

*We thank the reviewer for pointing out this error, there is only one correct value derived by Wooster et al. (2005) and that is 0.368 kg MJ-1, which is the value we used in this paper.*

**5. Page 7, Lines 22-29:**
Please see my comments concerning the interpretation of the "mean fuel load consumption" calculated using observations accumulated over long time periods (Page 8, Line 27 – Page 9, Line 2).

*Please see response to "specific comment 6" (below).*

**6. Page 8, Line 27 – Page 9, Line 2:**
Yes, I agree with the authors here. However I think they are overlooking a critically important aspect. Accumulating observations over long time periods (e.g., over many years) precludes a seasonal analysis. For the moment, consider a hypothetically static pre-burn fuel load that does not vary from the end of one rainy season to the beginning of the next rainy season. For a constant pre-burn fuel load that does not change over time, fuel load consumption will still vary depending on when during the dry season the landscape burns due to seasonal oscillations in fuel moisture contents, which drive seasonal oscillations in consumption completeness (Hoffa et al., 1999). Accumulating observations over long time frames fails to resolve the seasonal oscillations in consumption completeness and thus fuel load consumption. I strongly suggest that the authors warn the reader that estimates of fuel load consumption calculated from observations accumulated over long time periods are more representative of values observed at the peak in fire activity when the satellites detect the most active fire pixels and burned area pixels within a 0.25 degree grid cell.

Here's the really important bit though: the seasonality of fire activity is not always synchronized with the seasonality of fuel moisture contents and consumption completeness (Le Page et al., 2010). Hence peaks in fire activity may not always coincide with identical fuel moisture conditions. Across

the majority of Brazil, for example, the peak in fire activity generally occurs when fuels are driest (i.e., the middle of the fire season coincides with the middle of the CBI season, according to Figure 4 of Le Page et al., 2010). In contrast, across much of Africa, the middle of the fire season occurs before the middle of the CBI season. Therefore, even if the pre-burn fuel loads are identical between South America and Africa, the consumption completeness (%) at the time of peak fire activity would differ between the two locations due to differences in the seasonal synchronization of fire activity and fire weather, which would then lead to different estimates of fuel load consumption.

The authors do a nice job of using NPP and time since fire to explain geographical differences in the pre-burn fuel load, however they do not account for differences in consumption completeness (%), a value just as important in traditional estimates of fuel load consumption. In my opinion, the inability to identify fuel moisture conditions at the time of burning hinders a complete and confident interpretation of the geographical differences in fuel load consumption. Aligning the active fire and burned area pixels with a map of fuel moistures at the time of burning would be the ideal solution. However if the authors forgo such an analysis, they should at least make it extremely clear to the reader that fuel load consumption depends on the pre-burn fuel load AND consumption completeness, and that the latter is influenced by the environmental conditions and fuel moisture contents at the time of burning, which are not accounted for here.

*We agree that in addition to fuel loads fuel moisture plays an important role in the eventual fuel consumption by affecting the combustion completeness, and this likely explains part of the observed spatial variability in our long term mean fuel consumption estimate. However, in this manuscript much of our attention was focused on our main objective: "to derive mean fuel consumption estimates". We indeed carry out a first exploration of the possible drivers of the observed spatial patterns but limit ourselves to directly analyse the effect of NPP and fire return periods, partly because these can be relatively well observed from space and partly because they are expected to be key drivers of fuel consumption. Therefore it is only in the discussion that we mention the role of fuel humidity and combustion completeness on fuel consumption. A more detailed analysis, also including the temporal variation, would form an interesting follow up study. We have now expanded our discussion, to better highlight the importance of fuel moisture and combustion completeness and to discuss where largest effects may be expected and why (see updated sect. 5.2).*

*Page 22, lines 10 – 14 "Fuel consumption depends on the amount of fuel available for burning and the combustion completeness. In arid areas available fuel and thus fuel consumption is often limited by precipitation. Across these arid and semi-arid areas precipitation generally determines vegetation productivity and tree cover. Grasses in these more arid ecosystems often have a combustion completeness above 80% (van Leeuwen et al., 2014), and fuel consumption and fuel loads will generally be similar."*
*Page 23, lines 19 – 30 "For example, the highest fuel consumption in the more humid African savannas was found in the most frequently burning grid cells, suggesting a high combustion completeness. In areas where burning is largely limited by fuel humidity, the combustion completeness may have a considerable impact on fuel consumption. The fact that both frequently burning and almost fire free areas occur under similar climatic conditions in (sub)tropical savannas suggest that fuel conditions are important, while frequent fire occurrence may enhance flammability (Shea et al., 1996; Ward et al., 1996). Short fire return periods provide a competitive advantage to*

*herbaceous vegetation over woody vegetation (Bond et al., 2005; Bond, 2008). A high degree of canopy openness will result in more grass covered area and higher dry season ground surface temperatures and lower fuel moisture content resulting in high combustion completeness. However, a similar temperature or moisture driven effect may also be caused by the timing of the ignitions (Hoffa et al., 1999) directly related to management practices. Le Page et al. (2010) showed that African savannas typically burn early in the dry seasons, while Australian savannas often burn later in the season."*

**7. Page 10, Line 9 – Page 11, Line 14:**
Since the MCD64A1 product is used for both estimates, can it be assumed that the differences between FC MODIS and FC SEVIRI are entirely attributed to the different active fire products and the different methods for converting FRP to FRE?

*In the BGD paper it could in addition be explained by inter annual variation in fuel loads and combustion completeness since we compared MODIS derived fuel consumption (2003-2014) with SEVIRI derived fuel consumption (2010 – 2014). We have now updated Fig. 2 and compare MODIS and SEVIRI based fuel consumption over the same 2010 – 2014 time period. Therefore it can now be assumed that most differences indeed stem from sensor specific issues and different methods related to that, like the fire diurnal cycle in combination with the MODIS sampling design. Interestingly, little changes occurred in figure 2, indicating that although there may be diurnal, seasonal and inter annual variation in fuel consumption, the sample of 2010 – 2014 is reasonably similar to the longer term mean over 2003 – 2014. Note that to estimate $r^2$ (Fig. 2b) we exclude infrequently burning areas with burned area below 15% $yr^{-1}$, which may partly explain the good comparison using the different time periods.*

**8. Page 10, Line 9 – Page 11, Line 14:**
I'm curious about how much overlap there is in the fire activity that's driving the two estimates of fuel load consumption. I mean, are FC MODIS and FC SEVIRI driven by the same fire activity, or are they two different sets of fire activity, or is the fire activity that that drives FC SEVIRI a subset of the fire activity that drives FC MODIS? Based on the author's statement on Page 8, Line 21: "In contrast to the approach based on SEVIRI data, here all burned area observations were included", it would seem to me that FC SEVIRI is driven by fire activity that is a subset of the fire activity that is driving FC MODIS, if FC MODIS is limited to 2010-2014. Can any insights be gained by limiting the calculation of FC MODIS to the time period used to calculate FC SEVIRI (2010 – 2014)? Perhaps not the bias, but it seems to me that the scatter in the relationship between FC MODIS and FC SEVIRI could be due to the possibility that MODIS and SEVIRI are observing different fire activity within the same 0.25 grid cell. I may be mistaken, but I don't think this was ever offered as an explanation for the scatter. Unless the authors can demonstrate that estimates of FC MODIS and FC SEVIRI are driven by the same fire activity, then I think they have to concede that the scatter in the relationship could be attributed to the possibility that MODIS and SEVIRI are observing different fires within the same grid cell. Note that if fuel load consumption is homogeneous within a 0.25 degree grid cell, then it doesn't matter what fire activity MODIS and SEVIRI observed. However by their own admission, fuel load consumption is heterogeneous, and therefore has the potential to induce scatter in the relationship between FC MODIS and FC SEVIRI if MODIS and SEVIRI observe different fires within the same grid cell.

*In general it can be assumed that the MODIS instruments observe many "small" fires, that fall below the detection threshold of the SEVIRI instrument. However, the burned area product does not detect these small fires either, and "small" fires without detected burned area were excluded from the analysis in both the MODIS-derived and SEVIRI-derived fuel consumption estimates (Fig. 1). Moreover, we found that only 3% of annual burned area did not have corresponding SEVIRI FRP detections (Page 7 Lines 32-33). The sensitivity of the SEVIRI product to small fires is thus still be larger than that of the burned area product. Finally, especially in the case of the coarse SEVIRI grid cells (3x3 km at nadir) there is a small chance of small fires (not having burned area) burning alongside the larger fire within the same time window and grid cell. We expect that this effect on our fuel consumption estimates is small, as discussed in our response to "specific comment 3". Both datasets are thus largely based on the same fires (i.e., all the somewhat larger fires with burned area, except for the 3% excluded in the SEVIRI-approach). However, neither of the products (MODIS or SEVIRI) can observe the fire during its full life cycle, likely leading to most of the observed differences, as discussed below.*

*We have now made the comparison of SEVIRI and MODIS fuel consumption estimates based on the same time period (Fig. 2; 2010 – 2014). No major changes occurred in the figure and $r^2$ remains 0.42, indicating that most differences come from the actual sensor characteristics and related methods. We expect that most differences between the MODIS- and SEVIRI-derived fuel consumption estimates are caused by the different sensitivities of the MODIS and SEVIRI instruments and the fire diurnal cycle. Although the MODIS and SEVIRI instruments observe the same fires, as discussed above, they will observe those fires at different moments in time. The MODIS instruments only make observations at the fixed hours of their overpasses, while the SEVIRI instrument only observes the fire while FRP is above its much higher detection threshold. While peak daily fire activity of a fire large enough to leave a detectable burned area is nearly always observed by the SEVIRI instrument a certain fraction of daily fire activity will fall below its detection threshold. The relative fraction of FRE emitted below the detection threshold is likely a function of: (i) the SEVIRI pixel size, (ii) the shape of the fire diurnal cycle, and (iii) the size of the fire front and fuel consumption rate of the fire, that together drive absolute FRP values and thus SEVIRI's ability to detect the fire. In Fig. 2 two of these aspects clearly stand out:*

1. *The increasing SEVIRI pixel size and detection threshold away from the sub satellite point (0° N/S, 0° W/E) clearly lead to an underestimation of fuel consumption over southeast Africa.*
2. *In some areas with high fuel consumption SEVIRI and MODIS-derived fuel consumption estimates are nearly equal (i.e., red color in Fig. 2d), showing that nearly all fire activity occurred above the SEVIRI detection threshold. In areas with many small fires however, often only a relatively small fraction of the daily burning can be observed by the SEVIRI instrument and the MODIS estimates are thus considerably higher than the SEVIRI ones.*

*However, in order to fully resolve such issues a combined understanding of the influence of the fire diurnal cycle on both SEVIRI and MODIS FRE estimates is required. We made a start in Andela et al. (2015) to study the specific impact of the fire diurnal cycle on FRE estimates from MOSIS and to better characterize the fire diurnal cycle over Africa. But this is still an ongoing field of research.*

*In addition to the adjustments to Fig. 2, we have made several textual changes to reflect our current understanding of the differences between MODIS and SEVIRI-derived fuel consumption estimates:*

*Page 11 lines 12 - 22 "On top of these absolute differences, the spatial patterns were not uniform (Fig. 2b and d), for which we identified two main causes: first the MODIS based fuel consumption was consistently higher in south-eastern Africa (e.g., Mozambique and Madagascar), likely because of the decreasing sensitivity of the SEVIRI instrument at the greater off-nadir angle over this region* (e.g., Freeborn et al., 2014)*; and second the relative fraction of FRE emitted during periods that FRP values were below the SEVIRI detection threshold is a function of the absolute FRP values and the shape of the fire diurnal cycle. Fires with high FRP (related to high fire spread rates and/or fuel consumption) are often equally well observed by both instruments (i.e., red color in Fig. 2d), while areas with low fuel consumption are often characterized by a larger differences between the MODIS and SEVIRI estimates (i.e., green color in Fig. 2d)."*

*Page 20 lines 13 – 17 "We found that a large part of the differences could be attributed to the different sensors characteristics and methods used here. The shape of the fire diurnal cycle for example affects both MODIS based fuel consumption estimates due to the limited number of daily overpasses but also the SEVIRI derived fuel consumption estimates because it directly affects the relative fraction of daily fire activity that falls below the SEVIRI detection threshold."*

*Page 20 lines 24 – 29 "In our analysis a small part of the structural difference could also be explained by the fact that we did not correct for cloud cover and/or missing data in the SEVIRI based FC estimates. Not surprisingly, the best comparison between both methods was found in areas of high fuel consumption rates (Fig. 2d), for example areas where fires can spread over large areas to form large fire fronts (Archibald et al., 2013), and areas of high fuel consumption, these fires with high FRP are likely to be well observed by both instruments."*

*Page 21 lines 1 – 3 "Following previous studies, we find that about half of this discrepancy can be attributed to SEVIRI failing to detect the more weakly burning fires that ultimately are responsible for around half of the emitted FRE* (Freeborn et al., 2009, 2014)*."*

**9. Page 12, Lines 7 – 14:**
I think this section raises a very interesting question: is it reasonable to derive alternative conversion factors (kg MJ$^{-1}$) for different satellite sensors? To be honest, I don't know the answer to this question, so I leave it to the authors to pontificate. Note that the authors specifically state that: "Although the FRE per unit area burned can be converted to fuel consumption using the conversion factor found by Wooster et al. (2005) during laboratory experiments which we have used so far, some instrument specific issues may further affect the FRE estimates from space (see methods). In order to correct for uncertainties in the MODIS derived FRE estimates, we derived an alternative conversion factor by comparing the MODIS FRE per unit area burned directly to field measurements instead (Fig. 3a)." Here the authors admit that the reason for deriving an alternative conversion factor is because of the biases and uncertainties of estimating FRE from MODIS. If this is the case, then shouldn't the adjustment be more appropriately applied to the detection and conversion of FRP to FRE? Why should it be necessary to adjust an empirically derived "physical constant" due to satellite and sensor limitations? Are the authors suggesting that satellite sensor artefacts be incorporated into combustion chemistry and the relationship between radiant heat release and fuel consumption? Note that 0.356 kg MJ-1 (or 0.368 kgMJ-1) is a laboratory derived value, and as such its inverse represents (as close as possible) the amount of radiant heat released per kg of fuel

consumed. Adjusting this value to 0.572 kg MJ-1 means that its inverse has a different interpretation: the amount of radiant heat measured by MODIS per kg of fuel consumed. The difference in the meaning is subtle, but not trivial. For instance, there's a difference between (a) the radiant fraction, and (b) the fraction of total heat released that is measured as radiation by MODIS. The other option is to use the laboratory relationship between radiant heat and fuel consumption (which has a universal, physical meaning), and separately derive a MODIS specific FRP-FRE-adjustment factor to account for sensor limitations. I'll leave it to the authors to explain why deriving alternative, sensor specific conversion factors– as performed here – is a better option than using field measurements to derive an adjustment factor that's applied during the conversion of FRP to FRE. Although the two calibration options will obviously give you the same results, they nevertheless have different meanings.

*Our current understanding is that errors in the FRE estimation are indeed likely responsible for most of the difference between the two conversion factors. Considerable errors in absolute FRE estimates may for example be expected due to the MODIS sampling design in combination with the fire diurnal cycle (e.g., Vermote et al., 2009; Andela et al., 2015) and the increasing pixel size at higher scan angles (e.g., Freeborn et al., 2011). We appreciate the reviewers suggestion and have changed the manuscript accordingly. Rather than deriving an alternative conversion factor we now speak of a "FRE correction factor". In addition we followed the suggestion of reviewer #1 to use bootstrapping to get an estimate of the uncertainty involved with this correction factor (see manuscript with suggested changes and response to reviewer #1).*

**10. Page 15, Lines 8 – 20:**
To follow on from my previous comment, the authors recognize the role of productivity and fire return periods on the accumulation of fuels, and thus account for the impact of the pre-burn fuel load on fuel load consumption. However, the authors do not equally acknowledge the role of environmental and fuel moisture conditions at the time of burning and their influence on consumption completeness. Ideally the analysis should account for fuel moisture contents at the time of burning. If such an analysis is not feasible, then the authors should make every effort to remind the reader of the impact of different fuel moisture conditions on consumption completeness and thus fuel load reduction.

*We agree that fuel moisture (affecting the combustion completeness) is a major driver of spatiotemporal dynamics of fuel consumption, especially in the more humid tropics. We have made several textual adjustments to provide a more complete discussion of the role of fuel moisture and combustion completeness on fuel consumption. Please see our response to "specific comment 6" for further details.*

**11. Page 18, Lines 15-17:**
Couldn't the "large natural temporal variation in fuel consumption combined with the different periods of data availability" be discounted if FC MODIS and FC SEVIRI are compared between 2010-2014? Also, in addition to the different sensor characteristics, aren't there differences in the methods for converting MODIS and SEVIRI measurements of FRP to FRE (i.e., dividing by detection opportunities vs. temporal integration)?

*We appreciate this suggestion and now make the comparison (Fig. 2) using the same period (2010 – 2014) for both MODIS and SEVIRI derived fuel consumption. Interestingly, the difference between using MODIS-derived fuel consumption over the period 2010 – 2014 or MODIS-derived fuel consumption over the full study period (2003 – 2014) was small. Despite the temporal variation in fuel consumption the mean over 4 years seems representative for the full study period. For more details, please also see our response to "specific comments 7 and 8".*

*We agree that differences are both caused by the different sensor characteristics and the different methods to derive FRE per unit area burned from FRP and burned area observations. We have often referred to this as "differences in sensor characteristics", since it is for example the MODIS sampling design (a sensor characteristic) that forces us to choose an alternative way of deriving FRE as opposed to the continuous SEVIRI observations. However, since there are clearly alternative methods to estimate FRE using the MODIS observations it is indeed more correct to speak of "sensor characteristics and methods". We have updated the text accordingly.*

**12. Page 19, Line 30 – 34:**
This is one of only a few places where the authors disclose to the reader that fuel load consumption depends on the pre-burn fuel load and consumption completeness.

*Please see our response to "specific comment 6".*

**13. Page 20, Line 9 – 15:**
Indeed. Due to different management goals, fires are lit earlier in the dry season in Africa compared to South America, and particularly in Brazil where fires are generally lit at the peak of the dry season (as shown in Figure 4 of Le page et al., 2010). Hence fire management practices not only determine the fire return period, which affects the fuel load accumulation, but fire management practices also determine what time of year the fires are lit, and thus under what fuel moisture conditions the fires burn. Whilst the authors conclude that fuel load consumption across Africa is relatively low compared to Australia or South America due to differences in the fire return periods, one could also argue that based on the maps presented by Le Page et al. (2010), fuel load consumption is also lower in Africa since fires are more often lit earlier in the dry season when fuel moistures are higher and consumption completeness is lower.

*We now referred to the article by Le Page et al. (2010) and we have made several textual adjustments as highlighted in our response to "specific comment 6". Although the timing of the fires undoubtedly affects regional patterns of fuel consumption, especially in the more humid savannas, these patterns are also dependent on many other factors, like fuel loads and dry season duration. Our paper focused preliminary on the possibility of deriving fuel consumption estimates from satellite data while we provide a first exploration of the possible drivers of the spatial patterns. We hope that the new insights of this paper will provide the community with the tools to continue this research, getting to the actual underlying processes in more detail during follow up studies.*

**Technical Corrections:**

**1.** Page 1, Line 23-24: Incomplete sentence or incomplete thought.

*Now changed to "We used field measurements of fuel consumption to constrain our results, but the large variation of fuel consumption in both space and time complicated this comparison and absolute fuel consumption estimates remained more uncertain."*

**2.** Page 2, Line 19-21: Consider changing the sentence to read: "… is a key indicator of the consequences of changing management practices, vegetation characteristics and climate on fire regimes, as well as a key parameter required in fire emissions estimates."
*- Done*

**3.** Page 2, Line 21-23: Consider changing the sentence to read: "Yet, spatiotemporal dynamics of fuel consumption on a continental scale remain largely unmeasured and poorly understood (van Leeuwen et al., 2014)."
*- Done*

**4.** Page 3, Line 23: Consider changing the sentence to read: "… creating the first fully satellite-derived fuel consumption map for Africa."
*- Done*

**5.** Page 4, Line 1-2: Consider changing the sentence to read: "… in an attempt to provide more statistically representative fuel consumption estimates, particularly in less frequently burned grid cells.
*- Done*

**6.** Page 4, Line 7-8: Consider changing the sentence to read: "Finally, we used our fuel load reduction map to explore the drivers of fuel consumption in the study regions."
*- Not done, see our response to "specific comment 1".*

**7.** Page 4, Line 19-24: It may be helpful to remind the reader here that the MCD64A1 burned area product is also used by GFED.
*We now note this when we describe the GFED4s dataset:*
*Page 6 lines 23 - 25 "Methods used in GFED4s are based on GFED3.1 (van der Werf et al., 2010) but with two main improvements. The first one is the inclusion of small fire burned area in addition to the burned area observed by the MCD64A1 product (Randerson et al., 2012), the second .."*

**8.** Page 6, Line 24: Consider changing the sentence to read: "… first derived a fuel consumption map for Sub-Saharan Africa."
*- Done*

**9.** Page 8, Line 33: Grammar, pluralize: "Minimising the impact of these types of perturbations… "
*- Done*

**References**

[revised manuscript text omitted]